Corrected: Author correction

# New elevation data triple estimates of global vulnerability to sea-level rise and coastal flooding

Scott A. Kulp[1]* & Benjamin H. Strauss [1]

Most estimates of global mean sea-level rise this century fall below 2 m. This quantity is comparable to the positive vertical bias of the principle digital elevation model (DEM) used to assess global and national population exposures to extreme coastal water levels, NASA's SRTM. CoastalDEM is a new DEM utilizing neural networks to reduce SRTM error. Here we show – employing CoastalDEM—that 190 M people (150–250 M, 90% CI) currently occupy global land below projected high tide lines for 2100 under low carbon emissions, up from 110 M today, for a median increase of 80 M. These figures triple SRTM-based values. Under high emissions, CoastalDEM indicates up to 630 M people live on land below projected annual flood levels for 2100, and up to 340 M for mid-century, versus roughly 250 M at present. We estimate one billion people now occupy land less than 10 m above current high tide lines, including 230 M below 1 m.

[1] Climate Central, Palmer Square #402, Princeton, NJ 08542, USA. *email: skulp@climatecentral.org

Driven by climate change, global mean sea level rose 11–16 cm in the twentieth century[1,2]. Even with sharp, immediate cuts to carbon emissions, it could rise another 0.5 m this century[3–12]. Under higher emissions scenarios, twenty-first century rise may approach or in the extremes exceed 2 m in the case of early-onset Antarctic ice sheet instability[4,8]. Translating sea-level projections into potential exposure of population is critical for coastal planning and for assessing the benefits of climate mitigation, as well as the costs of failure to act.

Land topography and elevation, as represented by DEMs, lie at the foundation of such translation. High-accuracy DEMs derived from airborne lidar are freely available for the coastal United States, much of coastal Australia, and parts of Europe, but are lacking or unavailable in most of the rest of the world. By contrast, SRTM is a near-global satellite-based DEM covering latitudes from 56 south to 60 north and thereby land home to 99.7% of world population (based on 2010 Landscan data[13]). It is the standard choice for extreme coastal water level (ECWL) exposure analysis covering areas where high-quality elevation data are unavailable or prohibitively expensive[14–21].

SRTM models the elevation of upper surfaces and not bare earth terrain. It thus suffers from large error with a positive bias when used to represent terrain elevations. This is especially true in densely vegetated and in densely populated areas[22–25]. Mean error in SRTM's 1–20 m elevation band is 3.7 m in the US and 2.5 m in Australia when using DEMs from airborne lidar as ground truth[26]. Spaceborne lidar from NASA's ICESat satellite[27], a sparser, noisier and less reliable source of ground truth than airborne lidar, indicates SRTM has a global mean bias of 1.9 m in the same band[26]. This degree of error leads to large underestimates of ECWL exposure[28], and exceeds projected sea-level rise this century under almost any scenario[3–12].

In this article, we present ECWL exposure assessments that address this problem by employing CoastalDEM, a new DEM developed using a neural network to perform nonlinear, nonparametric regression analysis of SRTM error. This model incorporates 23 variables, including population and vegetation indices, and was trained using lidar-derived elevation data in the US as ground truth. CoastalDEM covers the same near-global latitudes as SRTM while reducing vertical bias to decimeter scale (0.01 m and 0.11 m as measured versus airborne lidar in the coastal US and Australia; −0.29 m as tested versus spaceborne lidar globally). CoastalDEM also cuts RMSE roughly in half compared to SRTM[26]. In low-elevation US coastal areas (where SRTM elevation is less than or equal to 20 m) in which population density exceeds 20,000 per square kilometer, including areas in 14 coastal cities such as Miami, New York City, and Boston, CoastalDEM reduces linear vertical bias from 4.71 m to less than 0.06 m. An overview of the methods used to generate CoastalDEM can be found in the methods section.

Central estimates in the recent literature broadly agree that global mean sea level is likely to rise 20–30 cm by 2050[3–10]. End-of-century projections diverge more, with typical central estimates ranging from 50–70 cm under representative concentration pathway (RCP) 4.5 and 70–100 cm under RCP 8.5[3,9,10,12], though more recent projections incorporating Antarctic ice sheet dynamics indicate that sea levels may rise 70–100 cm under RCP 4.5 and 100–180 cm under RCP 8.5, and could even exceed 2 m or more in far-tail scenarios[4,7,8,11]. Via a structured elicitation of opinion, experts now estimate there is a 5 percent chance 21st century sea-level rise will exceed 2 m[29]. Essentially all estimates are below the vertical bias of SRTM. Of these, we consider two representative sea-level projections for this assessment, labeled here as K14[3] and K17[4]. K14 is a probabilistic projection that is closely aligned with IPCC findings[10,30], while K17 is not probabilistic and emphasizes the possibility of more rapid sea-level

rise because of unstable ice-sheet dynamics[31]. Further details of these models are discussed in the methods section.

Both sets of projections are conditional on global carbon emissions; RCPs 2.6 (low emissions), 4.5 (moderate emissions), and 8.5 (high emissions) are considered for this analysis[32]. These models use 2000 as the baseline year (zero sea-level rise), which we treat as present-day with respect to sea level for relevant vulnerability estimates. The results we present here are based on median sea-level projections, along with 90% credible intervals when derived from K14, and 90% intervals from simulation frequency distributions when derived from K17 (we abbreviate both interval types as CI).

Because higher and more frequent coastal flooding is a direct impact of sea-level rise[33,34], we also assess potential exposure to ECWLs resulting from annual floods added on top of rising seas. We use local one-year return level heights (RL1) from the Global Tides and Surge Reanalysis[35]. These return levels vary spatially from a 5th percentile of 0.2 m to a 95th percentile of 2.8 m above local mean higher-high water (MHHW)—roughly speaking, the high tide line—across the near-global set of coastal cells assessed in this study (median value, 0.7 m).

We find that assessments using CoastalDEM instead of SRTM multiply median global ECWL exposure by roughly three or more for all scenarios and models considered. The majority of people living on implicated land are in developing countries across Asia, and chronic coastal flooding or permanent inundation threatens areas occupied by more than 10% of the current populations of nations including Bangladesh, Vietnam, and many Small Island Developing States (SIDS) by 2100.

## Results

**Global.** Given each sea level scenario analyzed (Supplementary Table 1), and alternately using SRTM and CoastalDEM, we estimate the number of people on land that may be exposed to coastal inundation—either by permanently falling below MHHW, or temporarily falling below the local annual flood height (Table 1, Supplementary Data 1). Coastal defenses are not considered, but hydrologic connectivity to the ocean is otherwise enforced using connected components analysis. Figure 1 presents permanent inundation surfaces at select locations for median K17/RCP 8.5/2100. Future population growth and migration are also not considered; rather, we use 2010 (essentially current) population density data from Landscan[13] to indicate threats relative to present development patterns.

Population exposure to projected sea level or coastal flooding is most commonly expressed as the total estimated exposure below a particular water level (total exposure)[14,16,17,19,21,36], but is increasingly also presented as the difference in exposure above a contemporary baseline (marginal exposure)[16,21,37]. Each approach has complementary strengths and limitations, discussed later. Here, we include marginal exposure values for key findings, while focusing more on total exposure. The latter is simpler and supports a wider and more easily interpretable set of comparisons between CoastalDEM-derived and SRTM-derived results.

For the present day, CoastalDEM estimates a global total of 110 M people on land below the current high tide line and 250 M on land below annual flood levels, in contrast with corresponding SRTM-based estimates of 28 M and 65 M. These values form the basis of the difference between total and marginal exposure estimates.

For one moderate future scenario, sea levels projected by 2050 are high enough to threaten land currently home to a total of 150 (140–170) million people to a future permanently below the high tide line, or a marginal increase of 40 (30–60) million. Total and marginal exposure each rise by another 50 (20–90)

**Table 1 Global populations on land at risk**

| Model | RCP | Frequency | 2050 | | | | 2100 | | | |
|---|---|---|---|---|---|---|---|---|---|---|
| | | | Total | | Marginal | | Total | | Marginal | |
| | | | CoastalDEM | SRTM | CoastalDEM | SRTM | CoastalDEM | SRTM | CoastalDEM | SRTM |
| K14 | 2.6 | Permanent | 150 (130–160) | 37 (32–43) | 40 (20–50) | 9 (4–15) | 190 (150–250) | 48 (36–65) | 80 (40–140) | 20 (8–37) |
| | | RL1 | 300 (270–320) | 78 (71–86) | 50 (20–70) | 13 (6–21) | 340 (300–400) | 95 (78–120) | 90 (50–150) | 30 (13–55) |
| | 4.5 | Permanent | 150 (140–170) | 38 (33–44) | 40 (30–60) | 10 (5–16) | 200 (160–260) | 52 (39–70) | 90 (50–150) | 24 (11–42) |
| | | RL1 | 300 (280–320) | 79 (72–87) | 50 (30–70) | 14 (7–22) | 360 (310–420) | 100 (82–130) | 110 (60–170) | 35 (17–65) |
| | 8.5 | Permanent | 150 (140–170) | 39 (33–45) | 40 (30–60) | 11 (5–17) | 230 (180–310) | 60 (44–85) | 120 (70–200) | 32 (16–57) |
| | | RL1 | 300 (280–330) | 80 (73–89) | 50 (30–80) | 15 (8–24) | 390 (330–460) | 110 (90–150) | 140 (80–210) | 45 (25–85) |
| K17 | 2.6 | Permanent | 150 (130–170) | 37 (29–47) | 40 (20–60) | 9 (1–19) | 190 (140–280) | 50 (32–75) | 80 (30–170) | 22 (4–47) |
| | | RL1 | 290 (260–330) | 77 (68–91) | 40 (10–80) | 12 (3–26) | 350 (280–430) | 97 (72–140) | 100 (30–180) | 32 (7–75) |
| | 4.5 | Permanent | 150 (130–180) | 38 (30–48) | 50 (10–80) | 10 (2–20) | 250 (170–380) | 64 (41–110) | 140 (60–270) | 36 (13–82) |
| | | RL1 | 300 (260–330) | 78 (68–92) | 40 (20–70) | 13 (3–27) | 400 (320–510) | 120 (84–180) | 150 (70–260) | 55 (19–120) |
| | 8.5 | Permanent | 150 (130–180) | 39 (31–50) | 40 (20–70) | 11 (3–22) | 340 (220–520) | 94 (56–180) | 230 (110–410) | 66 (28–152) |
| | | RL1 | 300 (270–340) | 81 (69–95) | 50 (20–90) | 16 (4–30) | 480 (380–630) | 170 (110–260) | 230 (130–380) | 105 (45–200) |

Estimates of people (millions) currently on land that may be exposed to permanent inundation or annual flooding (RL1) after projected (median and 90% CI) sea-level rise by the given year. Marginal values are included as well, estimating current occupants of land between present and projected future high tide and RL1. Population data from 2010 was used, when the estimated global total of population was roughly 6800 million

million people by end of century. A total of 360 (310–420) million people are on land threatened by annual flood events in 2100, or an extra 110 (60–170) million beyond the contemporary baseline. This case reflects greenhouse gas emissions cuts roughly consistent with warming of 2 °C (emissions scenario RCP 4.5) and assumes a mostly stable Antarctic (sea-level model K14).

In the case of Antarctic instability, a total of 300 (270–340) million people today live on land indicated as vulnerable to an annual flood event by mid-century, rising to as many as 480 (380–630) million by 2100. These values represent marginal increases of 50 (20–90) and 230 (130–380) million from the present, respectively. All 90% CIs given originate from uncertainty in sea-level projections.

More broadly, the effect on estimated ECWL exposure from changing the elevation data used exceeds the combined effects of emissions level, Antarctic behavior, and incorporation of annual flooding, as assessed using SRTM. For example, based on CoastalDEM, the total median current population on land falling below the projected mean higher high water line in 2100 under low emissions and a fairly stable Antarctica (RCP 2.6 and K14) is 190 million. This figure doubles the median SRTM-based estimate of 94 million under high emissions and Antarctic instability (RCP 8.5 and K17), and even exceeds SRTM-based figures under the same scenario after the addition of areas below the annual flood level (170 million).

More straightforwardly, Supplementary Data 2 and 3 tabulate people currently occupying land from 0–10 m MHHW at 1 m intervals, according to CoastalDEM and SRTM, respectively. In previous work using SRTM[18], about 640 M people have been estimated to live in the low elevation coastal zone (LECZ), defined as areas below 10 m. Defining the LECZ to reference MHHW instead of EGM96, we find SRTM predicts 780 M people below this threshold, and with CoastalDEM, the estimate rises to just over one billion people. Remarkably, this latter prediction includes 770 M below 5 m, versus 230 M from 5–10 m, illustrating a strong concentration in the lowest areas. The densest 1-m vertical band among the first ten is from 1-to-2 m, with 170 M inhabitants (or 1.7 M per vertical centimeter), pointing to a risky global pattern of development in light of sea-level rise.

**National**. With both SRTM and CoastalDEM, and regardless of emissions scenario or sea-level model, we find that more than 70% of the total number of people worldwide currently living on implicated land are in eight Asian countries: China, Bangladesh, India, Vietnam, Indonesia, Thailand, the Philippines, and Japan (Fig. 2, Supplementary Data 1). China alone accounts for 18–32% of global ECWL exposure across DEMs, depending upon the scenario, but CoastalDEM increases absolute estimates for China by a factor of roughly three compared to SRTM. Under K14/RCP 4.5, China could see land now home to a total of 43 (29–64) million people below MHHW by end of century, or 57 (30–100) million in the case of Antarctic instability (K17/RCP 4.5). The marginal increases in exposure from baseline are 20 (6–41) million and 34 (7–77 million), respectively. Under the same emissions scenario and either sea-level model, annual flood events at least double the corresponding estimates, threatening land occupied by over 60 million additional people.

In several developing countries south of China, ECWL exposure may be an order of magnitude more serious than previously expected as based on SRTM. As indicated by CoastalDEM, Bangladesh, India, and Vietnam come to rival China in the median number of people living on land implicated by 2100, totaling 21–30 million even under the low emissions

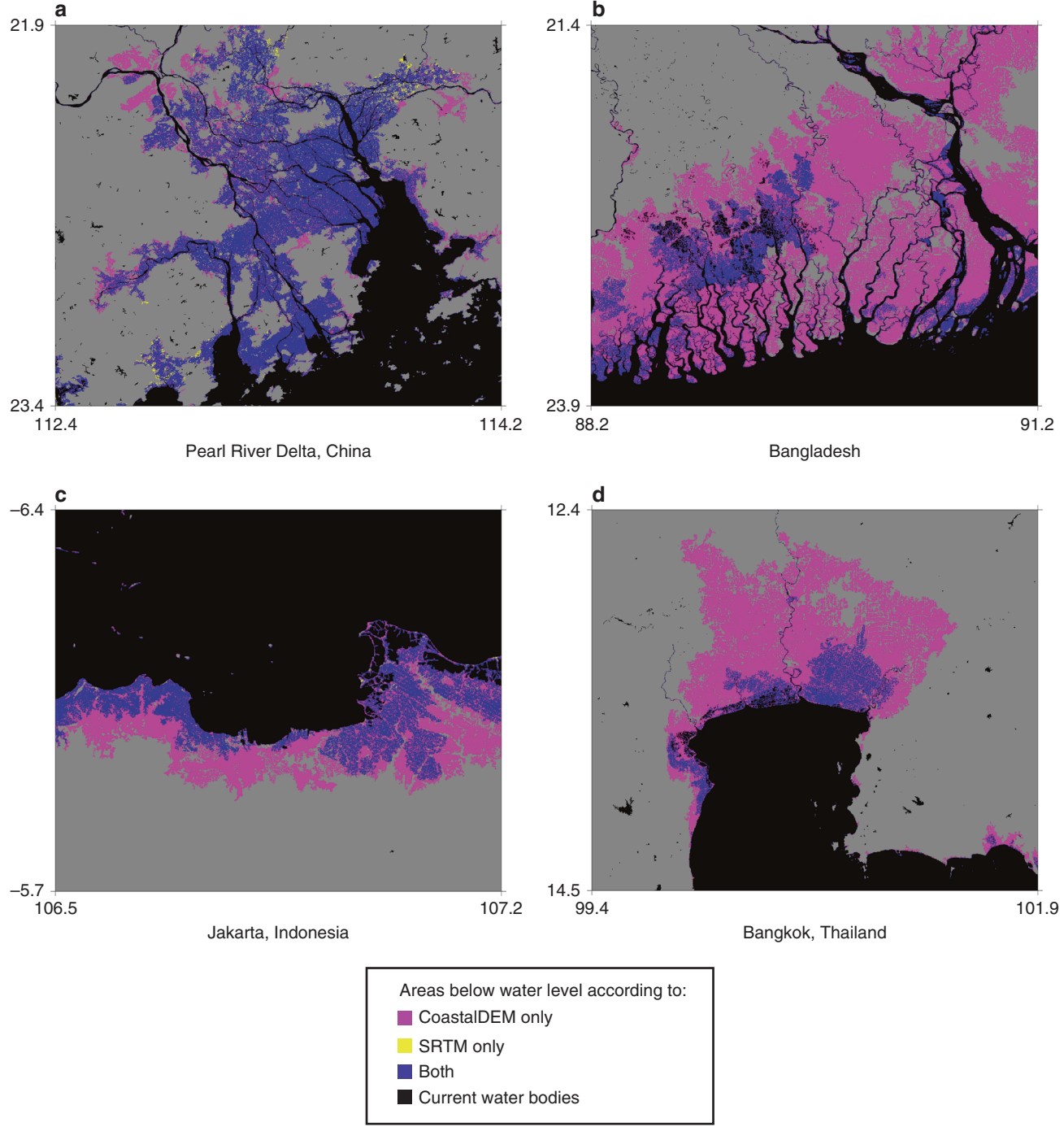

**Fig. 1** Permanent inundation surfaces predicted by CoastalDEM and SRTM given the median K17/RCP 8.5/2100 sea-level projection. Locations include (**a**) the Pearl River Delta, China; (**b**) Bangladesh; (**c**) Jakarta, Indonesia; and (**d**) Bangkok, Thailand. Low-lying areas isolated from the ocean are removed from the inundation surface using connected components analysis. Current water bodies are derived from the SRTM Water Body Dataset. Gray areas represent dry land. Axis labels denote latitude and longitude

scenario (K14/RCP 2.6), compared to 9–19 M today, and with another 7–20 million on land threatened by annual storm surge. Bangladesh, India, Indonesia, and the Philippines see a 5-fold to 10-fold change in estimated current populations below the projected high tide line after applying CoastalDEM. Globally, application of CoastalDEM leads to increased exposure estimates for the great majority of nations (Fig. 3).

Percentage rather than absolute exposure serves as a normalized metric of threat (Supplementary Data 4). In Asia,

CoastalDEM indicates that even with deep cuts to carbon emissions (K14/RCP 2.6), Bangladesh, Vietnam, and Thailand may, by end-of-century, face high tide lines higher than land now home to 19 (15–25)%, 26 (23–31)%, and 17 (15–18)% of their people, respectively, before accounting for episodic flooding events. These figures correspond to marginal exposure increases of 13 (9–19)%, 5 (2–10)%, and 15 (13–16)% of national populations. Continued high emissions with Antarctic instability (K17/RCP 8.5) could entail land currently home to roughly one-

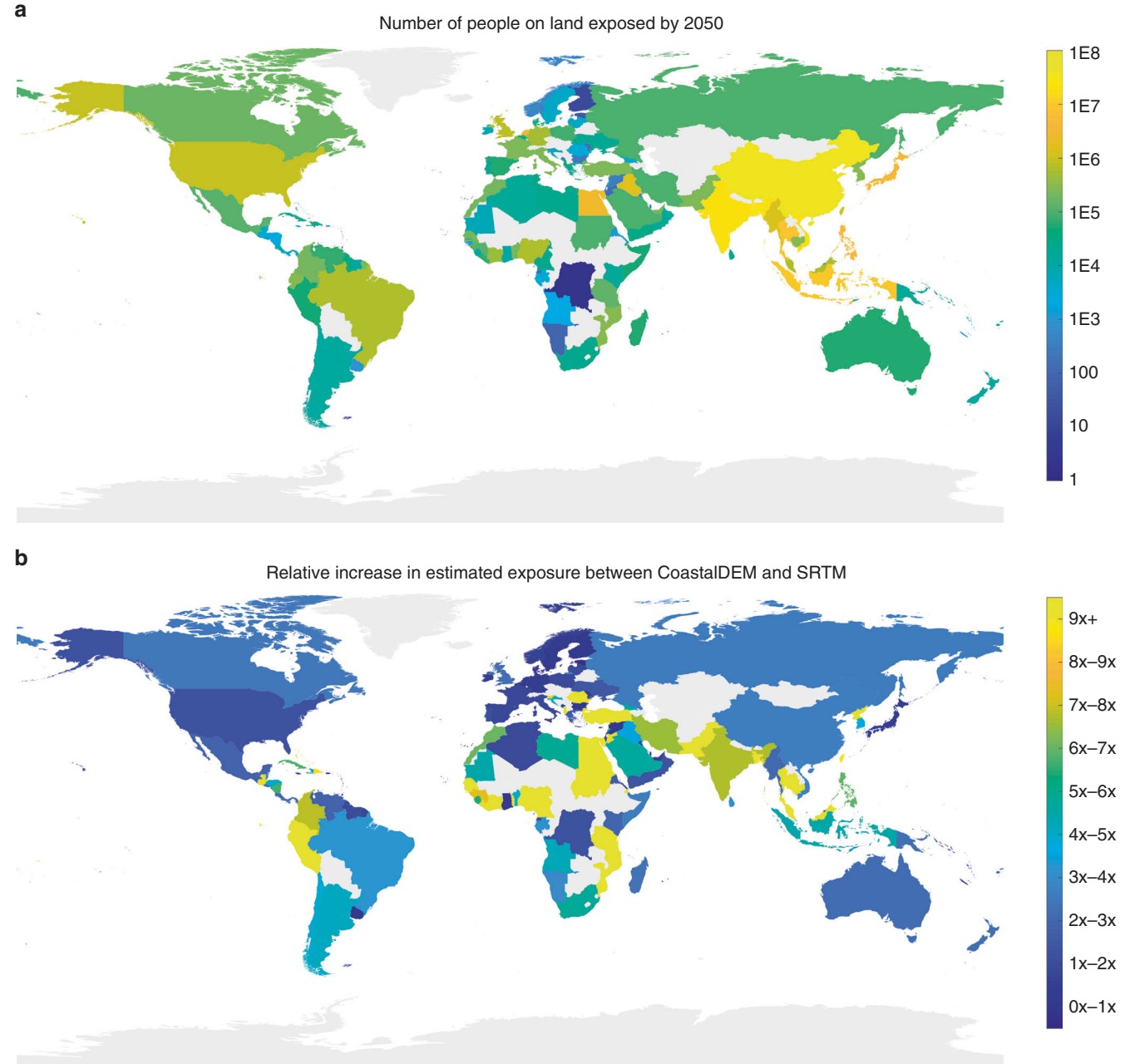

**Fig. 2** Total populations on vulnerable land. **a** Current population on land below projected mean higher high water level in 2100 assuming intermediate carbon emissions (RCP 4.5) and relatively stable Antarctic ice sheets (sea level model K14). Estimates based on CoastalDEM. **b** Factor by which CoastalDEM increases estimates of people on vulnerable land over SRTM in each country under K14/RCP 4.5. Countries wholly north of 60 degrees N are excluded because CoastalDEM is undefined at those latitudes. Source data are provided as a Source Data file. National boundaries based on public domain vector map data by Natural Earth (naturalearthdata.com)

third of Bangladesh's and Vietnam's populations permanently falling below the high tide line. It follows that some coastal municipalities within these nations will see even larger proportions of their populations threatened with displacement.

Outside of Asia and excluding the Netherlands, where an extensive flood control network is not captured by any of the elevation models studied, CoastalDEM indicates that 19 other countries are expected to see land currently home to 10% or more of their total populations fall below end-of-century high tide lines (based on median estimates), even under the deep emissions cuts of RCP 2.6. This count is up from two using SRTM. Except for

Djibouti and Guyana, all of these are island nations, and thirteen are classified by the United Nations as Small Island Developing States (SIDS).

Supplementary Data 1 and 4 provide results for the present, mid-century, and 2100.

**Validation.** The aspirational outcome of applying CoastalDEM to ECWL exposure analysis is to, as closely as possible, estimate the same amount of coastal vulnerability that a DEM derived from airborne lidar data would. We validate our results by first performing three representative ECWL exposure analyses using

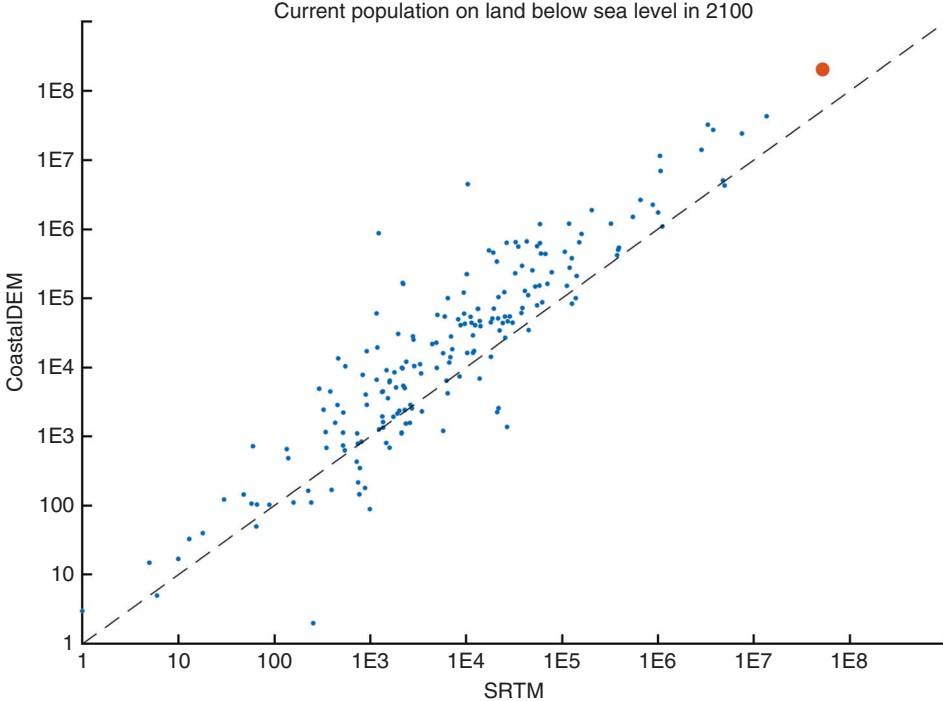

**Fig. 3** CoastalDEM versus SRTM by country. Each point represents a country, and its position corresponds to estimated total current population on land below the projected mean higher-high water level in 2100 (K14/RCP 4.5) using CoastalDEM (y-axis) versus SRTM (x-axis). The total global value is designated with the red point. Very large differences typically indicate large low-lying areas hydrologically connected to the ocean under CoastalDEM, but not SRTM. Source data are in Supplementary Data 1

lidar-derived data in the US and Australia. In Fig. 4, we plot the relative differences of predicted current population exposure between lidar and each global DEM at different water heights. Values of nearly zero imply a close match between exposure computed using both lidar and the target DEM, while larger absolute values suggest under-estimation or over-estimation of vulnerability. In addition to CoastalDEM and SRTM, we also include the alternative elevation models AW3D30 and MER-ITDEM, discussed more below.

We find that CoastalDEM strongly and consistently outperforms SRTM (as well as the other global DEMs) with this metric. At 1 m above MHHW, CoastalDEM improves linear relative difference in every state except for New York. Error is reduced from −69% (SRTM) to −43% (CoastalDEM) across the US, and from −77% (SRTM) to −23% (CoastalDEM) in Australia. Even larger improvements are seen at higher water levels, and at 3 m, relative errors in the US and Australia are smaller than −29 and 7%, respectively. We note that while the neural network that generated CoastalDEM was trained on lidar-derived data in the US, Australian lidar data is used only to validate the results, meaning strong results seen here mitigate fears that the model has been overfitted.

Error in the US is dominated by Florida, where an exceptionally large population occupies the coastal plain, and where SRTM vertical error in the southern half of the state is unusually high (exceeding 4 to 10 m). The neural network that generated CoastalDEM did not fully correct this large error. Discounting Florida, US relative error at 1 m drops from −62% (SRTM) to −30% (CoastalDEM)—a comparable improvement to that seen in Australia.

**Sensitivity analysis**. Spatial autocorrelation commonly characterizes DEM error, including error within SRTM[38]. SRTM error is strongly correlated with factors such as land slope[39], dense vegetation[24], and high population density[40], which themselves exhibit natural spatial autocorrelation. These features could manifest at any number of spatial scales (some towns may be only a few kilometers wide, while some urban agglomerations and forests are far larger). Furthermore, there exist well-known striping artifacts present in SRTM caused by satellite micro-adjustments[41], resulting, in cases, in multi-meter upward or downward bias across regions that may reach on the order of 100 km wide.

While CoastalDEM makes substantial improvements to SRTM, and includes, in its construction, inputs designed to reduce or eliminate striping, we anticipate that CoastalDEM also suffers from autocorrelated error. We therefore conduct a sensitivity analysis to explore the potential effects of error in CoastalDEM on our population exposure estimates, including the effects of autocorrelated error.

Monte Carlo simulations are regularly used to model DEM error and generate distributions of flood exposure estimates, from which uncertainty may be evaluated[38,42,43]. Such approaches typically either assume zero spatial autocorrelation, using the DEM's documented RMSE to generate random error surfaces[42,44]; or use low-pass filters across the error fields to simulate small-scale autocorrelation[45]; or employ sequential Gaussian simulations, which require widely dispersed ground-control-point data to accurately measure error statistics across the DEM[43,46]. The wide range of autocorrelation scale present here makes the second option unsuitable, and with no ground-control-point data available globally, the third is not possible.

Because of our expectations around the importance of spatial autocorrelation, we apply a modified, multi-scale approach to the first of these three methods. Assuming a normal distribution of error centered on zero and using a fixed global standard deviation, we generate 100 error fields using each of 6 different block sizes within which uniform error applies, ranging from 1

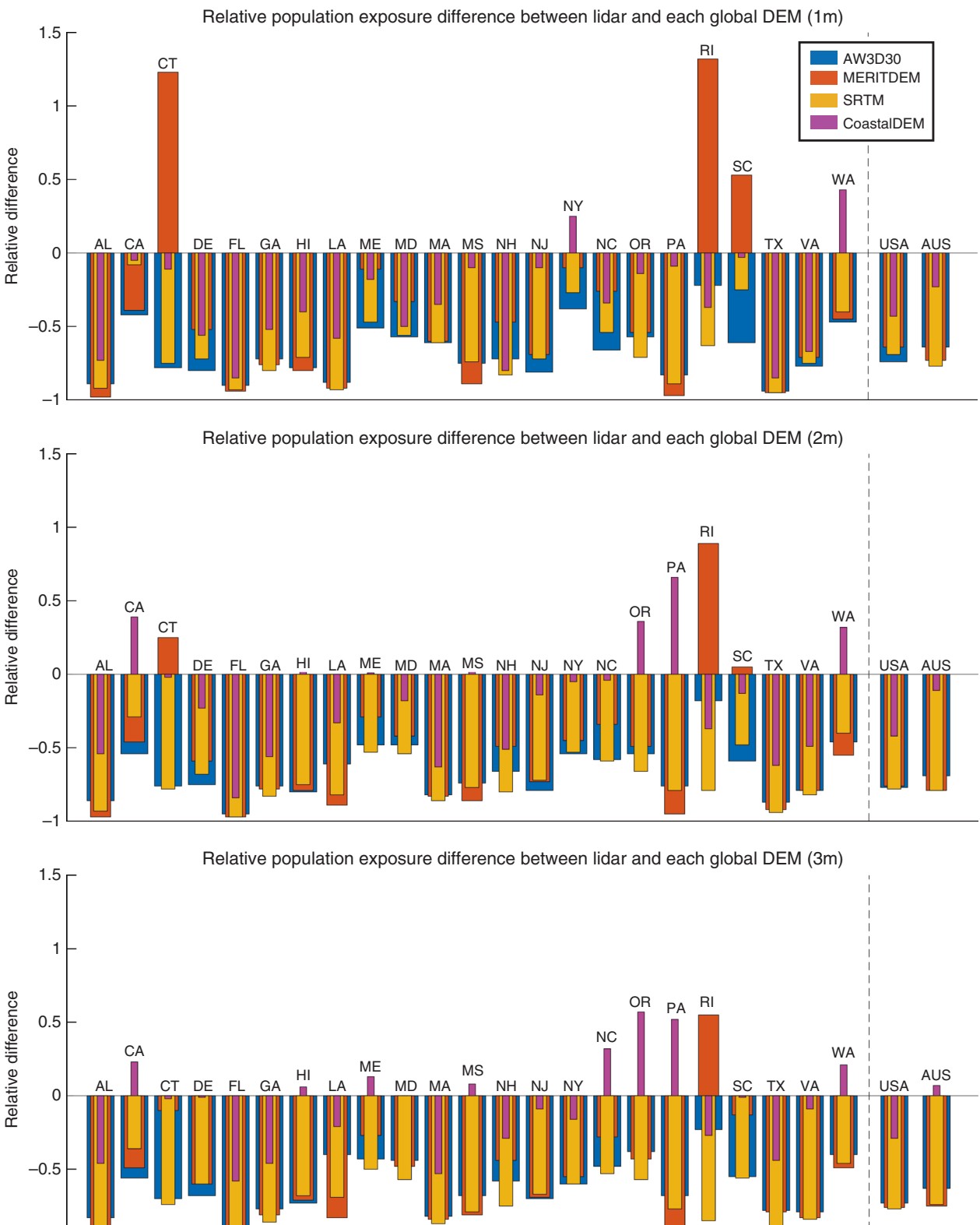

**Fig. 4** The relative difference of computed population ECWL exposure between lidar and four global DEMs. Populations living on land below 1, 2, and 3 m are computed in the US and Australia with each DEM. Zero relative differences indicate both lidar and the given global DEM predict the same number of people below the elevation threshold whereas, for example, −0.5 and 0.5 would indicate that the global DEM underestimation or overestimated by 50%, respectively. Results are given for each US state, as well as at the national scale in the US and Australia. Source data are provided as a Source Data file

## Table 2 Global simulated error assessment results

| Error resolution | Percentile | | |
|---|---|---|---|
| | 5th | 50th | 95th |
| Pixel (3 arcseconds) | 370 (−0%) | 370 | 370 (+0%) |
| 1 km | 370 (−0%) | 370 | 380 (+3%) |
| 0.1 deg | 380 (−3%) | 390 | 400 (+3%) |
| 0.25 deg | 380 (−3%) | 390 | 420 (+8%) |
| 0.5 deg | 370 (−8%) | 400 | 420 (+5%) |
| 1 deg | 360 (−10%) | 400 | 440 (+10%) |

100 simulated error surfaces are generated at each listed spatial resolution to represent different spatial scales of error autocorrelation, and added to CoastalDEM. 1-pixel simulations have no autocorrelation. Population exposure below 2 m local MHHW is computed for each simulated elevation dataset (each retaining CoastalDEM's original 3 arcsecond resolution), and the 5th/50th/95th percentiles of these results are presented. Percent differences from the median are provided for the 5th and 95th percentiles in parentheses. Units are in millions of people. Population exposure based on the unmodified CoastalDEM dataset is 400 M.

pixel (3 arcseconds) to 1 degree. We add the blocked errors to the original CoastalDEM to produce new simulated 3 arcsecond DEMs for computing exposure; the resulting exposure distributions are then evaluated separately for each block resolution. We use CoastalDEM's RMSE in Australia (2.46 m), as determined using lidar, to serve as the global standard deviation for our error distributions. We choose RMSE from Australia *versus* the US (RMSE 2.39 m) because the CoastalDEM model was trained in the US (albeit on just a 1% coastal sample). While vertical error will inevitably vary some from place to place, the similarity in error between the US and Australia increases our confidence in the value we employ.

We elect to use a water height of 2 m above MHHW (roughly and generally corresponding to a bad flood in the nearer term or an extreme sea-level scenario for 2100) as a case study. As in the main study, connected components analysis is used to remove isolated areas under the inundation surface before computing exposure. Unmodified CoastalDEM estimates 400 M people worldwide live below this threshold. Table 2 and Supplementary Data 5, respectively, provide global and country-level results for this sensitivity analysis.

Smaller error-block sizes (1-pixel through 1/10-degree resolution, roughly the size of a small city) produce highly consistent exposure estimates at the global scale, though biased low relative to the 400 M predicted without simulated error. This bias may be caused by higher spatial frequency DEM alterations cutting off some low-lying inland areas connected to the ocean through narrow pathways in the original CoastalDEM. Consistent with this mechanism, bias dissipates at larger error-block sizes. Also as autocorrelation scale grows, we see that 90% confidence intervals widen. At the extreme 1 degree resolution, roughly the scale of SRTM striping, the global 90% CI reaches plus or minus 10% about the 400 M median.

Countries also experience widening CI's across error resolutions, though considerably more rapidly than seen at the global scale. In countries with at least 1 M people below the 2 m threshold, the 90% CI's are, on average, plus or minus 2% about the median at 1 pixel, 5% at 1 km, 23% at 1/10 degree, 32% at 1/4 degree, 41% at 1/2 degree, and 49% at 1 degree. For example, at the 1-degree-error resolution, Bangladesh, India, and Vietnam have CI's of (−43 to 54%), (−40 to 27%), and (−29 to 23%) about their respective medians, while China is predictably less sensitive at (−21 to 21%). In general, larger areas of analysis and smaller error blocks lead to less sensitivity in ECWL exposure estimates, because each of these factors leads to larger random samples, making errors more likely to cancel out. Conversely, smaller areas and larger blocks each lead to smaller samples and more sensitivity.

These results suggest that CoastalDEM error exerts little influence on our global estimates, but reasonable caution should be applied when interpreting national scale assessments, particularly for smaller countries such as the SIDS. That said, we note that the 1-degree simulations represent worst-case scenarios, because they assume that CoastalDEM's RMSE derives exclusively from the largest considered spatial scale. Given the known factors at many spatial scales that contribute to DEM error, this assumption is unrealistic. Assessing characteristic error autocorrelation scales is beyond the scope of this study, but realistic CIs will be considerably narrower than implied by the 1-degree scale.

## Discussion

Despite improvements, elevation dataset error remains an important limitation in this study. We see that CoastalDEM still underestimates population exposure in both the US and Australia when compared to lidar-derived DEMs, suggesting the current assessment does not fully eliminate the bias in exposure estimates based on SRTM. CoastalDEM may still experience difficulty in dense cities, where exceptionally tall buildings in even the lowest-lying areas can cause SRTM elevations erroneously above 20 m. Since CoastalDEM is defined only where SRTM elevation is lower than or equal to 20 m, such areas are disregarded in this analysis, leading to some underestimation of exposure.

Older global scale DEMs, such as GLOBE[47] and GTOP030[48], have been used in previous work, and generally predict higher coastal flood exposure than SRTM[19,20]. However, their extremely high vertical errors (up to 100 m RMSE in both cases), low horizontal resolution (1 km), and spatial inconsistency in quality make them unreliable for ECWL vulnerability assessments. Their use for research has faded in comparison with SRTM, given its higher horizontal resolution and order-of-magnitude lower error. More recently, other DEM's have been released, such as AW3D30[49] and MERITDEM[50]. AW3D30 is a digital surface model primarily derived from stereo optical satellite imagery, and does not specifically attempt to improve vertical bias in either urban or forested areas. MERITDEM, like CoastalDEM, is based off of SRTM. It uses regression analysis to remove vertical error correlated with a number of vegetation metrics. However, MERITDEM does not seek to correct errors due to urban development. For sake of comparison, the analyses described in this article were repeated for these DEMs, and included in Supplementary Data 1 and 4. Results from both AW3D30 and MERITDEM, including US/Australia ECWL exposure error (Fig. 4), are generally consistent with those derived from SRTM, and so we maintain that these DEMs are equally inadequate for assessing coastal vulnerability.

Future modeling efforts may improve estimation of terrain elevations in tall-building districts and areas affected by SRTM striping. Ultimately, the most accurate assessments of vulnerability to rising seas, especially for smaller areas, will require development and public release of improved coastal area elevation datasets building directly off of new high resolution observations increasingly collected by satellites today.

Another limitation of this assessment comes from the population dataset Landscan, which is a 1 km$^2$ resolution model of ambient population density. While Landscan is widely used in the research literature, it cannot capture any bias toward or away from development within the lowest-lying coastal areas at sub-kilometer spatial scales. GRUMP is another population dataset with the same horizontal resolution, though it involves less sophisticated spatial modeling and is available only through 2000. It models nighttime (rather than ambient) population density[51], and has been shown to produce notably higher predictions of

exposure to ECWL[20]. Gridded Population of the World[52] is another alternative, based directly on census data without further modeling. While nominally at 1 km horizontal resolution, the data is piecewise constant between administrative boundaries, meaning its effective resolution is actually much coarser than Landscan. Newer datasets, such as Worldpop[53] and the High Resolution Settlement Layer[54], are anticipated to model population densities with higher accuracy at finer resolution, but are not yet available globally.

We emphasize that this analysis combines future water level projections with contemporary population densities. Results should therefore not be taken as projected impacts. Rather, they reflect the portion of presently developed land at risk in the future, which we interpret as a threat indicator. Efforts to integrate projected population growth, migration, economic development and coastal defenses into ECWL exposure projections have begun[19,36,55]. However, the spatial scales of socioeconomic projections remain very coarse compared to the scales at which elevation and current development data are available, posing a stiff challenge to their meaningful integration into analyses where fine-scale detail is critical. In addition, behavioral and economic responses to rising seas are likely to be unpredictable, due to the largely unprecedented nature and scale of the problem.

The vulnerability model employed in this analysis, a bathtub model where we classify all land below a given water height and hydrologically connected to the ocean as exposed to extreme coastal water levels, presents another partial limitation of the study. While this approach is reasonable in indicating land threatened with permanent inundation due to higher sea levels, it tends to overestimate exposure from episodic flooding, especially at small spatial scales[56,57]. It is likely that hydrodynamic models would predict less vulnerability to one-year floods than we estimate here. Areas accordingly misclassified as exposed to annual flooding would nonetheless likely face relatively frequent inundation risks.

Furthermore, this analysis assumes a static coastal topography, with the exception of a linear model of vertical land motion implicit in the sea-level projections used. Erosion, wetland migration/accretion, and other morphological processes are not considered. It is difficult to predict how these factors affect the uncertainty of our results, especially since sea-level change may trigger complex process responses. However, we note that armored, developed, and maintained shorelines in urban areas, where vulnerable populations are concentrated, may generally be less susceptible to such factors than undeveloped land.

This study focuses on estimating populations occupying land below future high tide lines or annual flood levels, but results also indicate that some 110 M people live below MHHW today (with many more below annual flood lines). Several explanations are possible. First, elevation error may drive the finding. However, in the US and Australia, CoastalDEM identifies fewer people living below MHHW (0.9 M and 69,000, respectively) than lidar-based analysis does (1.7 M and 75,000), consistent with our more general finding that CoastalDEM tends to underestimate coastal exposure relative to lidar.

Second, other sources of error may be important, including from the population data used and from the sea level data and tidal models used to determine local MHHW. A more detailed lidar-based analysis employing high resolution (block-level) US Census data[58] and NOAA's nearly continuous model for local MHHW[59] within the US cuts the original lidar-based estimate of 1.7 M nearly in half, to 0.9 M residents on land below MHHW. If these US results are indicative, and global population and MHHW estimates inflate exposure values derived from lidar elevation data, they likely also inflate values derived from CoastalDEM. Higher accuracy and higher resolution population,

sea level and tidal inputs are likely important for improving coastal exposure assessments in the future.

Third, many people today do in fact live on land below (or just above) MHHW, behind the protection of levees or other defenses. In the US, these account for 0.8 M out of the 0.9 M residents that our more detailed lidar analysis identifies as today occupying land below MHHW. Globally at present, levees and seawalls protect low-lying populations in many major deltas, such as around Shanghai, the Netherlands and New Orleans, and in areas experiencing rapid subsidence, such as parts of Jakarta and Tokyo. However, levee location data are not globally available, to our knowledge, and so are not incorporated into this analysis.

Fourth and finally, many people today do live in unprotected areas subject to frequent coastal flooding (if not below the high tide line), such as in Bangladesh, or in boats or structures on or above the water (such as homes on stilts). These possibilities are likely to be most common in developing countries, and to be poorly documented.

The levees, seawalls and other defenses and accommodations currently protecting tens or hundreds of millions of coastal-area residents globally point to the potential for protecting ever-larger areas as seas rise. At the same time, current coastal defenses should not be assumed adequate to protect against future sea levels and storms without continued maintenance and, likely, enhancement. These countervailing possibilities point to the merits of reporting results based both on total ECWL exposure and on marginal increases in exposure from the contemporary baseline. Total exposure recognizes the potential vulnerability of all populations on low-lying coastal lands as sea levels rise. Marginal exposure highlights new populations of concern, while leaving out populations in areas that may be defended at present, and thus may be more likely to be defended in the future.

Even in light of the limitations identified, this research, using a significantly improved model of coastal elevations, provides new best estimates of the vulnerability of populated low-lying areas to rising oceans at global and national scales. Reliability increases with the size of the area evaluated, and with the water level considered; thus, global assessments for end-of-century sea levels and floods, under high sea-level scenarios, should be considered most robust. Analysis reveals a developed global coastline three times more exposed to extreme coastal water levels than previously thought. Even with low carbon emissions and stable Antarctic ice sheets, leading to optimistically low future sea levels, we find that the global impacts of sea-level rise and coastal flooding this century will likely be far greater than indicated by the most pessimistic past analyses relying on SRTM. These results point to great need for the development and public release of improved terrain elevation datasets for coastal areas, for example via the high-resolution imagery and lidar point clouds increasingly collected by satellite today. There is also great need for improved population data; data on the location, height and condition of coastal-area levees and seawalls; and improved global sea-level and tidal models.

If our findings stand, coastal communities worldwide must prepare themselves for much more difficult futures than may be currently anticipated. Recent work has suggested that, even in the US, sea-level rise this century may induce large-scale migration away from unprotected coastlines, redistributing population density across the country and putting great pressure on inland areas[60]. It is difficult to extrapolate such projections and their impacts to more resource-constrained developing nations, though historically, large-scale migration events have posed serious challenges to political stability, driving conflict[61]. Further research on global-scale modeling of the timing, locations, and intensity of migratory responses to increased coastal flooding is urgently needed to minimize the potential human harm caused by such threats.

## Methods

**Sea-level projections.** We use two sea-level models for this assessment. K14[3] employs a probabilistic approach and includes very little contribution from Antarctica in its central projections. K17[4] links physical models of ice sheet loss to the projection framework established in K14, thus emphasizing the possibility of early-onset Antarctic instability[31]. However, the ice sheet model parameters used were not derived from probability distributions. Unlike K14, the resulting projection distributions produced by K17 are therefore considered simulation frequency distributions, rather than probabilistic ones. While more recent work[62] suggests that these Antarctic projections may be biased high, the resulting overall sea-level projections align roughly with the high end of what the sea level research community broadly expects[29]. Both models incorporate spatially explicit submodels for all climatic components of sea-level rise considered. They each also incorporate nonclimatic background contributions, such as glacial isostatic adjustment and sediment compaction. Leveraging sea-level records collected at 1022 PSMSL tide gauges worldwide, both K14 (updated in 2017[4]) and K17 employ a Gaussian process model to estimate nonclimatic contributions at points on a $2° \times 2°$ grid along the entire global coastline. Results for both models at the tide gauge and grid point locations are included in Supplementary Datasets 3 and 4 of Kopp et al.[4].

**CoastalDEM.** A multilayer perceptron (MLP) artificial neural network, a computational model often used for highly nonlinear non-parametric regression, was employed to predict the vertical error present at any SRTM pixel sample. MLP's are made up of layers of nodes in a weighted, directed graph, starting with an input layer (in our case, a 23-dimensional vector of known attributes at the target location) and ending with an output layer (1-dimensional error prediction at the target location). The neural network is trained by using lidar-derived elevation data in the US[63] as ground truth, iteratively adjusting weights in the graph to most accurately reproduce desired targets given the training set of samples. Our training set was made of over 51 million samples, and the 23 variables included neighborhood elevation values, land slope, population density, vegetation density, canopy height, and local SRTM deviations from ICESat altitude observations[64]. After training, the MLP predicted and removed SRTM errors at every pixel in the DEM with elevation between 1 and 20 m (inclusive). Details on the implementation and vertical error assessment of CoastalDEM were published earlier[26]. For this report, we used median resampling to convert CoastalDEM to a 3-arcsecond horizontal resolution.

**Vertical datum conversion.** We convert all elevation data to a common vertical reference frame (datum) for valid intercomparisons, electing the tidal datum mean higher high water (MHHW). MHHW is roughly equivalent to local high tide line and captures spatial variation in both mean sea level (MSL) and tidal amplitude. We use the globally extensive MSL model MSS_CNES_CLS_15[65], based on a 1993–2012 record of satellite sea surface height measurements from TOPEX/Poseidon, and referenced to the GLAS ellipsoid at 1-arcminute horizontal resolution. We also employ MHHW deviations from MSL provided by Mark Merrifield, University of Hawaii, developed using the model TPX08[66] at 2-arcminute horizontal resolution. Using NOAA's VDatum tool[59] version 3.7, we convert CoastalDEM, SRTM, AW3D30 and MERITDEM, plus the GLAS-referenced MHHW elevations, to a common ellipsoidal datum (WGS84). This allows us to subtract the elevation map of MHHW from each DEM to produce our final elevation maps above local MHHW.

A similar approach is taken in converting 1-year return levels to MHHW. The Global Tides and Surge Reanalysis, as distributed, is referenced to local MSL, so we use the MHHW-MSL deviation surfaces to change its vertical datum to MHHW.

**Exposure analysis.** Employing a modified bathtub model, we threshold each pixel in the DEMs to produce inundation surfaces at 0–10 m above MHHW. These inundation surfaces are computed at 1 m intervals with SRTM and AW3D30 (equivalent to their vertical resolutions), and at 0.25 m intervals with CoastalDEM and MERITDEM (which have continuous vertical resolutions). The surfaces are then refined using connected components analysis to remove all low-lying sub-threshold areas that the analysis indicates to be isolated by topography from the ocean.

To assess population exposure, we employ the LandScan 2010 High Resolution global Population Data Set, which estimates total populations living in 1 km$^2$ cells[13]. We refine this data using the SRTM Water Body Data Set, which defines land cells at up to 1-arcsecond resolution (30 m). We resample Landscan to align our DEM grids, assuming zero population in water cells, while proportionally increasing the population density in land cells 0–10 m above MHHW to ensure total population in each original 1 km square remains unchanged.

The population density grids are integrated under each 0.25 m-interval inundation surface, and tabulated according to the smallest administrative boundaries defined by the Global Administrative Areas (GADM) 2.0 Data Set[67]. In general, these administrative units are roughly the size of US counties or smaller. The local sea-level rise projections and 1-year return level heights, now referenced to local MHHW, are then computed and added at the centroid of each boundary by linearly interpolating from nearby sample points from the corresponding models. At the scale of these administrative units, the sea-level rise and RL1 gradients are relatively small, so any local factors affecting water heights are captured. Populations on land under each of these water heights are then estimated using

linear interpolation between the 0.25 m interval analyses. Results are aggregated to larger administrative areas, such as states and nations, as needed.

**Reporting summary.** Further information on research design is available in the Nature Research Reporting Summary linked to this article.

## Code availability

The methods described in this article were implemented using custom Matlab (R2017b), Python, and C++ code. Due to licensing restrictions by Climate Central, this code is not publicly available.

## Data availability

All exposure analyses (national populations on vulnerable land) that support the findings of our study are available within this article and its supplementary information files. The datasets SRTM, AW3D30, MERITDEM, Landscan 2010, and GADM are publicly available from their respective owners. The 3-arcsecond (90-m) version of CoastalDEM used in this analysis is available at no cost from Climate Central for non-commercial research use.

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

## Acknowledgements

The authors gratefully acknowledge Maya Buchanan, Michael Oppenheimer and Claudia Tebaldi for their thoughtful insights and comments on the manuscript. This research was supported by the Schmidt Family Foundation, the V. Kann Rasmussen Foundation, as well as One Earth, a project of Sustainable Markets Foundation, in partnership with the Leonardo DiCaprio Foundation, and grants from the National Science Foundation (ICER-1663807) and the National Aeronautics and Space Administration (80NSSC17K0698).

## Author contributions

S.K. and B.S. conceived and designed the analysis. S.K. performed the analysis. S.K. and B.S. wrote the paper.

## Competing interests

The authors declare no competing interests.
