## [Peer Review File · Nature Communications]

Reviewers' comments:

Reviewer #1 (Remarks to the Author):

1. Comments for Author

Thank you for the opportunity to evaluate this paper. The manuscript is very well motivated: the effects of climate change will have profound impacts on the planet, perhaps none more dramatically than coastal flooding due to sea level rise (SLR). Estimating the geographic extent of flooding impact on the human population is challenging. The research reported on here paper tackles this problem, in part, with a particular focus on the role of the measured elevation surface itself on the estimation.

The core objective is to use a new near-global digital elevation model (DEM) developed by the authors to assess the number of people around the planet in coastal areas likely to be affected by SLR in 2050 and 2100 using several SLR models and emissions scenarios. Results are contrasted with those using alternative global elevation products, and are summarized at both global and, in an appendix, national scales. Further, some mapped illustrative examples for small areas are included.

The most important finding is that, using the authors' higher-accuracy DEM, a much larger number of people are projected to be affected by SLR than by prior studies on less accurate DEMs. This result is defensible given what is known about the nature of errors in global DEM products, which tend to be biased above the actual ground level. It is important, as it offers a better way to quantify SLR impact and contributes to research on the costs of climate change. I therefore think the paper will be valuable and influential for a range of data and environmental scientists.

The paper is clearly written, with appropriate detail on background, methods, and results. Discussion is thoughtful and detailed, identifying strengths and, importantly, limitations of the work. A strength is the focus on DEM error and its impact on applications; most research implicitly assumes that the best available product is accurate and that application domain results are unbiased. This work demonstrates that the tendency of SRTM to overestimate surface elevations in vegetated and built-up areas has profound effects on the measurement of coastal inundation and flooding extent, and therefore the on magnitude of the affected population. One primary limitation that is discussed is that the work does not cover uncertainty in the 2010 gridded population estimates used to characterize exposure, or the uncertainty in future population size or distribution. As the paper notes, the results "indicate threats relative to present development patterns." Another limitation (also discussed in the paper) is that the work does not account for uncertainty in models of SLR. The use of multiple scenarios provides a sense of the range of this uncertainty. I am not a climate

scientist, but perhaps other reviewers could identify whether the models and scenarios chosen are the best or most representative to use.

My main concern with the paper is that uncertainty in the DEM is not really captured here. I agree that the authors' product, CoastalDEM, appears to be substantially better than SRTM (on which it is based) or other alternative DEM products. In their paper in *Remote Sensing of Environment*, the authors found that bias to 0 in the the US validation set and about 11 cm in the Australia validation set. RMSE, still the most commonly reported measure of error in DEMs, dropped from 4-5 m to around 2.4-2.5 m (Kulp & Strauss, 2018). These are real and substantial improvements. However, the magnitude of commonly anticipated errors remaining in the CoastalDEM product are meaningful: assuming that errors are normally distributed around zero with a standard deviation of 2.5 m, one third of all elevation estimates would be expected to have errors greater than 1 m above the true surface (and 18 percent 2 m or more above the true surface). These errors are greater than the anticipated amount of SLR in the 2050 and 2100 models, which range from 0.5 to 1.8 m. This implies that considerable uncertainty in the flooding extent remains in CoastalDEM due to data error, and this injects considerable uncertainty into the estimation of global population at risk.

One solution is to discuss the problem. Another would be to develop a model - perhaps a very simple model - and simulate error on CoastalDEM, then calculate flooding for each simulation, and then calculate population at risk for each simulation, thereby constructing a distribution of population estimates equal to the number of simulations. A third approach would be to conduct simulation on a few illustrative case studies. My suspicion is that the results would indicate wide confidence intervals around the estimate of population at risk, and in particular that the number of people directly affected by flooding due to SLR could be substantially greater even then the estimates calculated in the paper.

Finally, I have a few comments about the figures. Figures and tables, including in the supplement, contribute to the work. Figure 1 may not be interpretable by some color-blind readers, as it includes brown and green. Graphic resolution may need to be increased in some or all figures for paper publication. Figure 3 includes two maps of all countries on the earth. The map is unprojected, which leads to large amounts of spatial distortion. As a geographer, I would urge the authors to use projected data (maybe Robinson, Winkel, Goode homolosine, anything but Peters!) for a stronger presentation of their global data. Important aspects of Figure 4 are not clear to me, either from the graphic, the caption, or the in-manuscript discussion. What does "error in estimates of population exposure below X_m " mean? What to the differences between 1, 2, and 3 m imply? What are the units of the y-axis?

Overall I enjoyed the paper. It is a clearly communicated and carefully researched piece of scholarship with important implications for the future of coastal areas. Its focus on the quality of

global data, which form the basis of so much big science, is also an important lesson, as those data are critical to the quality of the output of that science.

Reference

Kulp, S.A, and Strauss, B. H. (2018). CoastalDEM: A global coastal digital elevation model improved from SRTM using a neural network. *Remote Sensing of Environment* 2016: 231-239.

Signed,

Ashton Shortridge

Professor, Department of Geography

Michigan State University, USA

Reviewer #2 (Remarks to the Author):

This work uses a recently released coastal digital elevation model, CoastalDEM, to estimate land and population exposure by mid and late 21st century as a consequence of projected sea level rise and for elevations below 20 m above present-day mean sea level. The digital elevation model has been developed by the same authors and is described in detail and compared to other sources of data in Kulp and Strauss (2016) (<https://doi.org/10.1016/j.rse.2017.12.026>). Projections of mean sea level and extreme sea levels use the updated state of the art values.

One of the obvious important applications of a global coastal DEM is the estimation of coastal areas vulnerable to projected sea level rise and marine extreme episodes. In this sense, the authors have made a good job to exploit the potential value of their product. The manuscript is mostly clear (exceptions listed below); it is also concise and quantifies the impact of an improved DEM for the range of projections at the global and national levels in terms of population exposure. The authors discuss the limitation of the present study, the most important of which are the remaining errors in the DEM and the use of present-day population data without any projection. Overall, I think the paper can be a relevant contribution that highlights the large uncertainties in one of the costliest impacts of global warming.

In my opinion, there are two relevant items that should be better addressed in this work. First, uncertainties in CoastalDEM have been quantified comparing with ICESat in the former paper by the

authors. Despite the limitations of ICESat measurements there are some numbers along the world coastlines (figure 6 in Kulp and Strauss, 2016) that can be used to estimate the errors in the exposure assessment. I think this could be more interesting than the comparison with other global DEMs presented in Tables S2 and S3, as it would provide some bounds on the reliability of the results.

Second, coastal topography and coastlines are assumed not to change in response to sea level rise, which might be accurate in urban areas, but not that much is sedimentary coasts that are very common. It is not only that the bathtub approach does not work, as pointed out by the authors, the problem is that morphological processes are much more complex even not accounting for extreme sea levels, only in response to long-term sea level rise. This discussion (in page 11) should be extended in this respect.

Figures need to be improved to make the legends, axes and ticks easily readable. In Figure 1, the present-day coastline would be useful and would serve as a guide to the interpretation of the results. Figure 4 needs a better description. In this figure 4 different models are compared, but only 2 of them are discussed in the validation section, when the figure is first cited (the other 2 are discussed later). So, the caption should be more informative. As it is now I am really confused of what is being shown here.

Some other (minor) comments:

Line 58: local sea level projections are referenced here, but the values are referred to table S1 in which only global (not local) mean sea level rise is listed. Thus, it is unclear whether the authors have used regionalised sea level rise scenarios, provided by Kopp et al (2014, 2017) or the global values. The correct option is the former.

Line 82: how is hydraulic connectivity considered?

As a final comment I would like to note that the fact that the CoastalDEM data is not publicly available could be problematic in terms of the repeatability of the results. However, I will assume that this has to do with the journal data policy rather than with a scientific review.

Reviewer #3 (Remarks to the Author):

Review of 'New Elevation Data Triple Estimates of Global Vulnerability to Sea-level Rise and Coastal Flooding' by Scott A. Kulp and Benjamin H. Strauss

General remarks

In this manuscript, the authors combine CoastalDEM, an improved Digital Elevation Model (DEM), with a simple bathtub model to assess the number of people that live in areas that will be inundated during the 21st century due to projected sea-level rise. They find that, depending on the sea-level rise scenario, the number of people at risk is substantially larger than found in previous estimates.

This is an interesting and relevant topic, and accurate DEMs can be a helpful tool for decision makers to get a grasp of the problems at play. However, in my feeling, the manuscript and the methodology are not yet in a mature stage, and some substantial work is needed before the work can be published. I'm a bit afraid that this paper is going to generate big headlines, despite the significant, but unquantified uncertainties in the used methodology that are likely too large to substantiate the presented numbers. I have four major points that should be addressed before the paper is in a publishable form.

1. The uncertainties related to the used methodology are discussed, but not adequately quantified. Since the uncertainties of the methodology are rather substantial, a quantitative analysis is needed to assess the validity of the presented numbers. For example, in the worst-case scenario, the authors find that up to 640 million people live in areas at risk. How significant is this number? Is it somewhere between 630 and 650 million people, or somewhere between 100 million and a billion people? Now, the only uncertainty that is quantified is the sea-level projection uncertainty, while I have the impression that the biases and uncertainties in the DEM are much larger than the projection uncertainty. This quantification is especially important given the large numbers of people affected and the consequences of flooding.

2. Furthermore, the sea-level projections are taken from Kopp et al., 2014 (K17) and 2017 (K17). While these projections are robust, they consist of projections at individual tide-gauge locations, and contain estimates of local subsidence rates. In this manuscript, these numbers are averaged by country, which is used as the projected sea-level change for the country. The K14 and K17 projections include local land motion processes, based on individual tide-gauge records, which are now averaged over whole countries.

For countries with large coastlines and few gauges or coastlines for which large spatial variations in the expected rise exist, this extrapolation will cause substantial errors. An example of such an issue is the US: The Eastern coastline will see a much larger sea-level rise than its Western coast, due to GIA and ocean dynamic effects. However, in this study, both coasts are assumed to see the same mean sea level rise.

A similar issue occurs with local land subsidence: depending on whether the tide gauges are affected or not, this process is or is not included in this analysis.

To avoid these issues, I suggest to use gridded projections, such as presented in the IPCC AR5 scenarios, which cover the global oceans. These projections allow a clear separation between the large-scale sea-level rise (included) versus local subsidence (excluded), and they ensure that the regional patterns are preserved.

Another issue that should be made clear is that the estimates for the Antarctic Ice Sheet contribution in K17 are based on DeConto and Pollard, 2016, whose projections are considered biased high by some people in the cryosphere community, see for example Edwards et al. 2019. While discussing these issues in detail is of course far beyond the scope of this manuscript, it may be worthwhile to clearly state that the numbers in Kopp et al. 2017 are at the high end of the spectrum, even among studies that take ice dynamical instabilities into account.

3. Another point that would be helpful to discuss is how many people are already living below the annual flood level, based on the the DEMs. Are the numbers presented in this work the extra people below this level, or is it the total number of people in areas at risk?

4. Finally, there is room for extra figures, and I think these figures can be helpful to show some representative flooding levels in cities, which show the effect of various scenarios and different DEM approaches (Airborne Lidar, ICESAT, SRTM and CoastalDEM), so we get a grasp of the relative importance of these effects on local level. Furthermore, maps with population density susceptible to flooding for various scenarios are welcome as well. Now there is only one map which only shows the impact on a country-by-country level.

Line-by-line comments

L14-15: I don't see where the claim about human consequences is substantiated in the paper. I suggest to remove this sentence. See also the comments about L250 ff.

L64: what are 'simulation frequency distributions', and what is its impact on the projected uncertainty?

L76: As far as I understand, you separate two different height levels: annual flooding and MHHW. Is 'permanent inundation' the same as 'below MHHW'?

L230: The conclusions drawn here about more difficult futures, migration pressure etc do not follow from the presented results. Although I agree that this is a common-sense claim, linking changes in

flood levels to economic damage, migration patterns, and human suffering is a whole science on its own, and I'd stay away from conclusions that cannot be directly drawn from the paper. I suggest to remove this part of the conclusions.

L250: I'm a bit confused with the reference levels: you refer all DEMs to present-day MHHW using a global tide model. How do you combine this model with GTSR? Do you derive MHHW from GTSR and subtract this number from total water levels to derive annual exceedance levels?

L280: I encourage the authors to publish the data tables in computer-readable form, such as text tables or Excel sheets, as well as the used DEM, in a public repository. In my experience, this lowers the bar for many people to use the results of this study.

References

Edwards, T. L., Brandon, M. A., Durand, G., Edwards, N. R., Golledge, N. R., Holden, P. B., ... & Wernecke, A. (2019). Revisiting Antarctic ice loss due to marine ice-cliff instability. *Nature*, 566(7742), 58.

Reviewer #1 (Remarks to the Author):

1. Comments for Author

My main concern with the paper is that uncertainty in the DEM is not really captured here. I agree that the authors' product, CoastalDEM, appears to be substantially better than SRTM (on which it is based) or other alternative DEM products. In their paper in Remote Sensing of Environment, the authors found that bias to 0 in the the US validation set and about 11 cm in the Australia validation set. RMSE, still the most commonly reported measure of error in DEMs, dropped from 4-5 m to around 2.4-2.5 m (Kulp & Strauss, 2018). These are real and substantial improvements. However, the magnitude of commonly anticipated errors remaining in the CoastalDEM product are meaningful: assuming that errors are normally distributed around zero with a standard deviation of 2.5 m, one third of all elevation estimates would be expected to have errors greater than 1 m above the true surface (and 18 percent 2 m or more above the true surface). These errors are greater than the anticipated amount of SLR in the 2050 and 2100 models, which range from 0.5 to 1.8 m. This implies that considerable uncertainty in the flooding extent remains in CoastalDEM due to data error, and this injects considerable uncertainty into the estimation of global population at risk.

One solution is to discuss the problem. Another would be to develop a model - perhaps a very simple model - and simulate error on CoastalDEM, then calculate flooding for each simulation, and then calculate population at risk for each simulation, thereby constructing a distribution of population estimates equal to the number of simulations. A third approach would be to conduct simulation on a few illustrative case studies. My suspicion is that the results would indicate wide confidence intervals around the estimate of population at risk, and in particular that the number of people directly affected by flooding due to SLR could be substantially greater even then the estimates calculated in the paper.

This is an important point, and one raised by both of the other reviewers. In response, we have added a supplementary discussion, where we, as you have suggested, generated a number of simulated error surfaces, applied them to CoastalDEM, and assessed population on land at risk. This is an extremely computationally intensive exercise, though, so we elected to do this analysis on Australia alone, where we already know its RMSE. Since DEM error tends to be spatially autocorrelated due to factors such as vegetation, urban development, and striping artifacts, we repeated this experiment using a number of different error surface horizontal resolutions to model different degrees of autocorrelation.

Finally, I have a few comments about the figures. Figures and tables, including in the supplement, contribute to the work. Figure 1 may not be interpretable by some color-blind readers, as it includes brown and green. Graphic resolution may need to be increased in some or all figures for paper publication. Figure 3 includes two maps of all countries on the earth. The map is unprojected, which leads to large amounts of spatial distortion. As a geographer, I would urge the authors to use projected data (maybe Robinson, Winkel, Goode homolosine, anything but Peters!) for a stronger presentation of their global data. Important aspects of Figure 4 are not clear to me, either from the graphic, the caption, or the in-manuscript discussion. What does "error in estimates of population exposure below Xm" mean? What to the differences between 1, 2, and 3 m imply? What are the units of the y-axis?

We recognize the reviewer's concern about the colors Figure 1. We note, though, that the importance of this figure is the contrast between water and land, and not the exact elevation and/or the brown to green color scale, so most color-blind readers should have little or no difficulty interpreting the core results presented in this image. Additionally, and unfortunately, creating this figure was very time-intensive, so we strongly prefer to keep the figure with its original colors.

Figure resolution has been improved. Figure 3 is now projected, and Figure 4 (now Figure 5 in the revised manuscript) has been changed (both in the main text, the images, and the caption) to better clarify these results.

Reviewer #2 (Remarks to the Author):

In my opinion, there are two relevant items that should be better addressed in this work. First, uncertainties in CoastalDEM have been quantified comparing with ICESat in the former paper by the authors. Despite the limitations of ICESat measurements there are some numbers along the world coastlines (figure 6 in Kulp and Strauss, 2016) that can be used to estimate the errors in the exposure assessment. I think this could be more interesting than the comparison with other global DEMs presented in Tables S2 and S3, as it would provide some bounds on the reliability of the results.

ICESat measurements are very sparse and relatively low quality; compared to airborne lidar, we had previously found that ICESat is biased over a meter too high in the coastal zone in the US.

That said, Reviewer 1 had similar concerns regarding uncertainty of the results, and as described above, we added a supplementary discussion to address this.

Second, coastal topography and coastlines are assumed not to change in response to sea level rise, which might be accurate in urban areas, but not that much is sedimentary coasts that are very common. It is not only that the bathtub approach does not work, as pointed out by the authors, the problem is that morphological processes are much more complex even not accounting for extreme sea levels, only in response to long-term sea level rise. This discussion (in page 11) should be extended in this respect.

We have added a paragraph to the discussion about this concern.

Figures need to be improved to make the legends, axes and ticks easily readable. In Figure 1, the present-day coastline would be useful and would serve as a guide to the interpretation of the results. Figure 4 needs a better description. In this figure 4 different models are compared, but only 2 of them are discussed in the validation section, when the figure is first cited (the other 2 are discussed later). So, the caption should be more informative. As it is now I am really confused of what is being shown here.

We have increased font sizes to improve legibility.

We appreciate the reviewer's comments about figure 1, but believe adding the present-day coastline would create too much visual complication. The main point of the figure is to give a visual contrast between results from lidar and the other DEMs, which we believe it accomplishes.

Figures have been re-rendered at a significantly higher resolution to make them clearer. As mentioned in an above comment to the first reviewer, Figure 4 (now Fig 5) has been reworked to be made more readable.

Line 58: local sea level projections are referenced here, but the values are referred to table S1 in which only global (not local) mean sea level rise is listed. Thus, it is unclear whether the authors have used regionalised sea level rise scenarios, provided by Kopp et al (2014, 2017) or the global values. The correct option is the former.

We use local relative sea level projections, discussed in more detail in the methods section. The global values provide only reference points for discussion. We have clarified this in the reference to Supplementary Table 1.

Line 82: how is hydraulic connectivity considered?

We use connected components analysis to remove isolated low-lying areas. This is discussed in more detail in the methods section, but has also been clarified in the main text.

As a final comment I would like to note that the fact that the CoastalDEM data is not publicly available could be problematic in terms of the repeatability of the results. However, I will assume that this has to do with the journal data policy rather than with a scientific review.

The 90m version of CoastalDEM, which is the same version used in this paper, will be available at no cost for noncommercial research use. This is noted in the Code/Data Availability section.

Reviewer #3 (Remarks to the Author):

1. The uncertainties related to the used methodology are discussed, but not adequately quantified. Since the uncertainties of the methodology are rather substantial, a quantitative analysis is needed to assess the validity of the presented numbers. For example, in the worst-case scenario, the authors find that up to 640 million people live in areas at risk. How significant is this number? Is it somewhere between 630 and 650 million people, or somewhere between 100 million and a billion people? Now, the only uncertainty that is quantified is the sea-level projection uncertainty, while I have the impression that the biases and uncertainties in the DEM are much larger than the projection uncertainty. This quantification is especially important given the large numbers of people affected and the consequences of flooding.

We have further addressed uncertainty quantification of CoastalDEM -- see the discussion with reviewer 1 and the supplementary discussion we have added to the submission.

2. Furthermore, the sea-level projections are taken from Kopp et al., 2014 (K17) and 2017 (K17). While these projections are robust, they consist of projections at individual tide-gauge locations, and contain estimates of local subsidence rates. In this manuscript, these numbers are averaged by country, which is used as the projected sea-level change for the country. The K14 and K17 projections include local land motion processes, based on individual tide-gauge records, which are now averaged over whole countries. For countries with large coastlines and few gauges or coastlines for while large spatial variations in the expected rise exist, this extrapolation will cause substantial errors. An example of such an issue is the US: The Eastern coastline will see a much larger sea-level

rise than its Western coast, due to GIA and ocean dynamic effects. However, in this study, both coasts are assumed to see the same mean sea level rise.

A similar issue occurs with local land subsidence: depending on whether the tide gauges are affected or not, this process is or is not included in this analysis.

To avoid these issues, I suggest to use gridded projections, such as presented in the IPCC AR5 scenarios, which cover the global oceans. These projections allow a clear separation between the large-scale sea-level rise (included) versus local subsidence (excluded), and they ensure that the regional patterns are preserved.

We respectfully note that the reviewer has misunderstood. We are not averaging sea-level rise projections across countries. Rather, we are using nearest-neighbor values from the 2-degree gridded K14 (updated in the Kopp 2017 paper) and K17 projections to determine local relative sea-level rise at each level-2 unit from GADM (a global administrative boundary database), the equivalent of US counties. We perform the exposure analysis at each unit individually, and sum them to the national scales. This way, we capture the relevant local factors. This has been clarified in the methods section of the manuscript.

Another issue that should be made clear is that the estimates for the Antarctic Ice Sheet contribution in K17 are based on DeConto and Pollard, 2016, whose projections are considered biased high by some people in the cryosphere community, see for example Edwards et al. 2019. While discussing these issues in detail is of course far beyond the scope of this manuscript, it may be worthwhile to clearly state that the numbers in Kopp et al. 2017 are at the high end of the spectrum, even among studies that take ice dynamical instabilities into account.

We have clarified this in the manuscript.

3. Another point that would be helpful to discuss is how many people are already living below the annual flood level, based on the the DEMs. Are the numbers presented in this work the extra people below this level, or is it the total number of people in areas at risk?

The numbers reflect the total number of people in areas at risk. We have added a note on the present-day estimate (280M).

4. Finally, there is room for extra figures, and I think these figures can be helpful to show some representative flooding levels in cities, which show the effect of various scenarios and different DEM approaches (Airborne Lidar, ICESAT,SRTM and CoastalDEM), so we get a grasp of the relative importance of these effects on local level. Furthermore, maps with

population density susceptible to flooding for various scenarios are welcome as well. Now there is only one map which only shows the impact on a country-by-country level.

We already have an inundation surface comparison between lidar, SRTM, and CoastalDEM in Figure 1, and ICESat is not actually a DEM, but rather a very noisy and sparse set of points on a set of linear tracks going around the world. That said, in these revisions, we have added a figure presenting inundation surfaces for four select high-density locations worldwide, using SRTM and CoastalDEM.

In our 2018 paper in RSE (“CoastalDEM: A global coastal digital elevation model improved from SRTM using a neural network”), we performed extensive analysis on the impacts of population density on both SRTM error and the resulting CoastalDEM output, so we feel that similar figures here would not substantially add to the discussion.

Line-by-line comments

L14-15: I don't see where the claim about human consequences is substantiated in the paper. I suggest to remove this sentence. See also the comments about L250 ff.

This sentence has been removed from the abstract.

L64: what are ‘simulation frequency distributions’, and what is its impact on the projected uncertainty?

This idea is detailed in Kopp et al 2017. In brief, the sea level projections draw upon an Antarctic ice sheet model for which a certain set of parameter values was explored, in order to cover a reasonable range of potential values for each parameter. However, the researchers could not assign a probability to any parameter value. The resulting simulations therefore present a distribution of values from which percentiles may be taken (simulation frequency distributions), but do not necessarily correspond to probability distributions.

L76: As far as I understand, you separate two different height levels: annual flooding and MHHW. Is ‘permanent inundation’ the same as ‘below MHHW’?

Correct. We have clarified this in the manuscript.

L230: The conclusions drawn here about more difficult futures, migration pressure etc do not follow from the presented results. Although I agree that this is a common-sense claim,

linking changes in flood levels to economic damage, migration patterns, and human suffering is a whole science on its own, and I'd stay away from conclusions that cannot be directly drawn from the paper. I suggest to remove this part of the conclusions.

We are not making any quantitative conclusions on damages and so on. Instead, we are highlighting, as you say, the common-sense implications of our results, and why this all matters the wider scientific community, policy makers, and the general public. We maintain that this section is an important part of the article, so at this time we have not removed it.

L250: I'm a bit confused with the reference levels: you refer all DEMs to present-day MHHW using a global tide model. How do you combine this model with GTSR? Do you derive MHHW from GTSR and subtract this number from total water levels to derive annual exceedance levels?

GTSR, as distributed, is referenced to mean sea level. We convert this to MHHW before adding these heights to the SLR projections at the centroid of each GADM unit, and then computing exposure under the same. This has been clarified in the methods section of the manuscript..

L280: I encourage the authors to publish the data tables in computer-readable form, such as text tables or Excel sheets, as well as the used DEM, in a public repository. In my experience, this lowers the bar for many people to use the results of this study.

We have added CSV data files as supplementary materials for this article. The 90m version of CoastalDEM (the same used in this study) will be available at no cost for noncommercial research use following publication of this paper.

References

Edwards, T. L., Brandon, M. A., Durand, G., Edwards, N. R., Golledge, N. R., Holden, P. B., ... & Wernecke, A. (2019). Revisiting Antarctic ice loss due to marine ice-cliff instability. *Nature*, 566(7742), 58.

Reviewers' comments:

Reviewer #1 (Remarks to the Author):

The authors have made substantial changes that have improved the manuscript. I think it makes a substantial and important contribution. My advice to the editor is that some more does need to be done. Specific comments about some key responses follow (Rx-y: x indicates Reviewer 1, 2, or 3, while y indicates the question from that reviewer).

R1-1) Simulation modeling to study impact of error.

I appreciated the decision to implement this and recognize the computational challenges of doing so. Conducting it on several small areas is a reasonable choice. The authors are correct that spatial autocorrelation needs to be accounted for in the error model. Their approach generates noise realizations matching the standard deviation of observed error characteristics (this is a common approach) at a range of cell sizes (this is atypical). The use of ranges of cell sizes is intended to capture aspects of spatial autocorrelation that could affect sensitivity of cells to SLR, but it may miss out on other aspects. I'm generally ok with it, but if a reference to this simulation methodology is possible, that would strengthen support for this approach.

Results from this model are striking - in particular, considerably more populated areas appear to be at risk when accounting for error, even though the error model assumes no bias in the direction of elevation errors.

Some text issues in this supplemental material:

"severe underestimation seen in Florida (Figure 4)" - I think Figure 5?

"probably due to a large second derivative of population exposure in Australia as a function of water height around 2 m" - The meaning is unclear. Is there an increase in population counts for an elevation of ~2m ASL in Australia?

"presented in Figure ??." - I think Figure 4?

R1-2, R2-3, R3-4) Figure improvements.

The figures are better.

I still don't like Figure 1, though I both appreciate its point (most readers have no idea how serious errors in global DEMs can be) and also why the authors don't want to change it (computational expense). However, standard factors like the areal extent of each tile, local context (e.g., current coastline) and absolute location (lat-lon axes) are missing which detracts from the utility of this figure for readers.

Figure 2 (added due to R3-4) is valuable. This figure could use scale bars in each figure (or just once if scales are equivalent) so readers know how large the differences between CoastalDEM and SRTM are. This figure would also benefit from adding lat-lon axes.

The projected world maps are a big improvement.

I think the new Figure 5 is better than the old one.

R2-2) Bathtub model oversimplifies impact of SLR on coastlines.

A short paragraph (line 232) has been added to address this. It's clear modeling of geomorphological processes that affect coastline position is beyond the scope of this paper, but it is worth noting that the impact of SLR on how those processes operate is unknown and could also play a major role in affecting coastal population vulnerability (and could work to increase population vulnerability in some areas while decreasing it in others), particularly in non-urban areas.

Ashton Shortridge

Reviewer #2 (Remarks to the Author):

This new version is only slightly revised and changed with respect to the original submission, despite some major concerns pointed out by three different reviewers. My recommendation therefore remains the same: the manuscript should still undergo a revision before being published.

One major issue in the previous version was the poor treatment and discussion of the uncertainties in CoastalDEM (only uncertainties associated to mean sea level rise were accounted for, but not

uncertainties in the DEM product). This is a point that has been raised by the three reviewers. To address this, the authors have performed an additional exercise to evaluate how uncertainties in CoastalDEM are translated into uncertainties in exposure, using Australia as a case study. I see two problems with the response: firstly, it is now hidden in a file of the S.I. while, due to its relevance, it should be more central to the discussion and highlighted in the main text. Secondly, and perhaps the reason why it is in S.I., I cannot draw a main conclusion from the sensitivity tests. Does it serve to estimate an uncertainty range in population exposure associated to the use of this DEM? The use of multiple spatial resolutions in the DEM fields is clearly a limitation. I do not think that this is the way it should be done; instead, I think that the DEM fields constructed using uncertainties should have the same resolution as the original CoastalDEM and should be spatially correlated. This is a non-trivial task and I am not suggesting that the authors follow this approach. An alternative, easier way to bound the uncertainty in the population exposure associated to the errors in CoastalDEM would be to increase/decrease the DEM by the same relative amount everywhere around the Australian coast: for example, +/- 50% of the RMSE at every pixel.

Regarding the figures:

- Modification in Figure 1: I suggested to include the present-day coastlines in Figure 1 as the locations are extremely difficult to recognise without this information. I think no one could actually identify the coast of Tampa using only flooded Lidar data shown in the first column of the figure. I understand from the authors' responses that this may be a computationally costly action, but I believe this is needed to properly interpret the figure.

- Figure 5: there has been an improvement in the quality, but I still find the caption insufficient to interpret what is shown here. Why are now values centred at -0.5? what is exactly the y-axis? In the former version I would have said that it represents a percentage, but now I am unsure given its average value.

- Line 84: this sentence needs to be rewritten: if there is permanent flooding below MHHW, then is not due to ESL but to mean sea level rise. Only temporary flooding is associated to ESL.

It would have been very helpful that the authors indicated where (e.g. page, lines) the new modified text has been included, or, even better, would have provided a file with tracked changes.

Reviewer #3 (Remarks to the Author):

Review of 'New Elevation Data Triple Estimates of Global Vulnerability to Sea-level Rise and Coastal Flooding' by Scott A. Kulp and Benjamin H. Strauss, round 2

General remarks

The updated manuscript is a substantial improvement over the previous version, although there are still some open issues that were raised in the previous round, but have not yet been fully addressed in my opinion.

The major weakness of this paper still forms the estimates of the uncertainty. Although I can see that a full assessment of all sources of uncertainty is not feasible to perform, the readers are still left in the dark on how to assess its uncertainty on a global scale, at least after reading the supplement, I still don't have an idea of how to interpret the numbers in the main text. The authors have added a case study in Australia, but some basic guidelines or a first-order estimate how these local results translate into other regions and the global-mean case is still missing.

Furthermore, I still don't understand how the K14/K17 projections are interpolated on the whole coastline. These projections are not gridded but computed at a set of tide-gauge locations. This is especially important given the background rates that are included in K14 (Figure 6c), which are derived from local tide-gauge data and not from gridded estimates. Are these local values interpolated along the coastlines? Do you also interpolate the values from Figure 6c? This is especially important, given that this term contains large scale (e.g. GIA) and local (e.g. subsidence due to groundwater pumping) effects, which are not always suitable for interpolation. For reproducibility, the followed procedure should be part of the methods section.

Finally, in the previous review, I asked about the number of people currently living in risky areas. While this number has been added, I wonder where does the number '280 million' come from? From coastalDEM? In that respect, is there a difference in the number of people who are now not inundated and will be in the future between SRTM and coastalDEM? Both the present-day and future inundation are biased low in SRTM I guess. Another option would be to express all numbers as the change in number of affected people relative to the present-day baseline. Furthermore, this high number of people already under the annual flood level would put the large numbers of people affected in better perspective. The maximum number of people at risk is 640 million, which is just more than double the number of people presently at risk. I therefore strongly suggest to make this number of 280 million very clear from the beginning, to avoid the risks that this paper generates news headlines that are not justified. Finally, how many people nowadays live below MHHW? Many coastal cities in Europe and Asia have large parts protected by seawalls.

Line-by-line remarks

L7: A recent paper by Gregory et al. (2019) suggests a standardization of many sea-level related terms. One of them is 'Extreme Coastal Water Level' instead of Extreme Sea Level. I'd like to encourage the authors to consistently use the terminology proposed by Gregory et al (2019)

L11; 'More than 200 million'. From Table 1, I get a value of 'more than 170 million', which is less than 3 times the value of 65 million from SRTM.

L14: 360M mid-century: This number is not discussed in the main text. I suggest to remove it.

L45: Square mile: Please use SI units.

L95/ Table 1: Where do these uncertainties come from? It looks like they're based on the projected uncertainties only, but that should be made explicit.

L105FF: This is a good example on why the baseline would matter. How high are these numbers today, and do they depend on SRTM vs coastalDEM?

L132-L137. I already raised this point in the previous section, but I still don't see the added value of these statements, as these effects are already known, and the probability of these effects are not further quantified in this manuscript. This is a scientific paper after all, and not a textbook or press release.

L248-L249: These statements only hold for regions and communities that have based their expectations on SRTM or other unreliable DEMs. Is this actually the case? I suspect that many places either have used a high-resolution estimate (developed countries) or do not have any estimate at all (under-developed countries). About the immigration pressure: this is not trivial either, see for example Eseban et al, 2018, who show that people tend to stand their ground. I again suggest to stick to the computed numbers in the conclusions.

References:

Esteban, M., Jameró, Ma. L., Nurse, L., Yamamoto, L., Takagi, H., Thao, N. D., ... Shibayama, T. (2019). Adaptation to sea level rise on low coral islands: Lessons from recent events. *Ocean & Coastal Management*, 168, 35–40. <https://doi.org/10.1016/j.ocecoaman.2018.10.031>

Gregory, J. M., Griffies, S. M., Hughes, C. W., Lowe, J. A., Church, J. A., Fukimori, I., ... van de Wal, R. S. W. (2019). Concepts and Terminology for Sea Level: Mean, Variability and Change, Both Local and Global. *Surveys in Geophysics*. <https://doi.org/10.1007/s10712-019-09525-z>

We would again like to thank the editor and reviewers for their time and thoughtful comments. We have used these suggestions to substantially improve this manuscript.

The most notable of these changes is in the sensitivity analysis. While the methodology remains broadly the same, this analysis has been expanded to full global scale, and also applied for each individual coastal country in the world. The results of this approach not only reinforce the reliability of our global estimates, but also reveal limitations when assessing vulnerability within smaller countries. Additionally, we note that this updated analysis has been added to the main manuscript, rather than being included as supplementary material, as requested by one reviewer.

We have also computed global- and national-scale estimates of present-day population counts below current MHHW and the 1-year return level. The results are discussed in the main manuscript, along with new discussions on potential sources of error regarding these measurements. Estimates for each country are included in the supplementary materials.

In the time between revisions, we have also found minor software issues that resulted in the connected components analyses occasionally missing certain isolated areas in the CoastalDEM-derived inundation surfaces (this did not affect SRTM or other DEMs). This has been corrected and we have updated the manuscript accordingly. At global scale, this corresponds to about 10-20M fewer people on vulnerable land, which does not meaningfully change the conclusions nor impact of this report.

Finally, we have made changes based on smaller reviewer comments; we have made a number of other minor editorial adjustments for clarity based upon our own review; and we have added very brief discussion related to simple exposure estimates for population on land from 0-10 m MHHW, which in our view adds impact to the manuscript.

We have attached two versions of the revised manuscript -- the first is a clean copy with no tracked changes, and the second highlights where most of the changes were made. Due to technical difficulties generating these notations within latex, a very few changes are not highlighted. These include the deletion of the former Figure 1; the addition of Table 2; the addition of a new section on sensitivity analysis; the removal of the former Supplementary section on sensitivity analysis; changes to Supplementary tables; addition of several references; changes in reference numbering; and text within figures and the caption of Table 1. We believe this is a complete list.

Point-by-point responses to the reviewers follow.

Reviewer #1 (Remarks to the Author):

The authors have made substantial changes that have improved the manuscript. I think it makes a substantial and important contribution. My advice to the editor is that some

more does need to be done. Specific comments about some key responses follow (Rx-y: x indicates Reviewer 1, 2, or 3, while y indicates the question from that reviewer).

R1-1) Simulation modeling to study impact of error.

I appreciated the decision to implement this and recognize the computational challenges of doing so. Conducting it on several small areas is a reasonable choice. The authors are correct that spatial autocorrelation needs to be accounted for in the error model. Their approach generates noise realizations matching the standard deviation of observed error characteristics (this is a common approach) at a range of cell sizes (this is atypical). The use of ranges of cell sizes is intended to capture aspects of spatial autocorrelation that could affect sensitivity of cells to SLR, but it may miss out on other aspects. I'm generally ok with it, but if a reference to this simulation methodology is possible, that would strengthen support for this approach.

In the revised manuscript, we provide a short overview of the literature on this topic. Monte Carlo simulations of DEM error and their impact on coastal vulnerability assessments have been done in past work. The reviewer is correct, though, that some particulars of our methodology (the range of error resolutions to better understand the effect of autocorrelation) are unique. In previous research where autocorrelation is taken into account, it's generally either by applying a small (generally a $\sim 3 \times 3$ pixel) filter to the error surface to blur it and simulate autocorrelation (not applicable at the scales we are considering), or by incorporating ground control point data to assess local error statistics across the DEM (not possible without GCP data).

"severe underestimation seen in Florida (Figure 4)" - I think Figure 5?

"probably due to a large second derivative of population exposure in Australia as a function of water height around 2 m" - The meaning is unclear. Is there an increase in population counts for an elevation of ~ 2 m ASL in Australia?

"presented in Figure ??." - I think Figure 4?

The sensitivity analysis has been substantially revised to the extent that these issues are no longer present.

I still don't like Figure 1, though I both appreciate its point (most readers have no idea how serious errors in global DEMs can be) and also why the authors don't want to change it (computational expense). However, standard factors like the areal extent of each tile, local context (e.g., current coastline) and absolute location (lat-lon axes) are missing which detracts from the utility of this figure for readers.

While we feel the 3D renderings are powerful imagery, upon further thought and consideration, we have come to the conclusion that this figure is redundant with the inclusion of the former Figure 2 (now Figure 1, in the current submission, comparing inundation surfaces between CoastalDEM and SRTM), and given that Figures 3 and 4 of (Kulp & Strauss, 2017) already provide direct elevation comparisons between lidar, SRTM, and CoastalDEM. We have therefore removed this figure from the manuscript.

Figure 2 (added due to R3-4) is valuable. This figure could use scale bars in each figure (or just once if scales are equivalent) so readers know how large the differences between CoastalDEM and SRTM are. This figure would also benefit from adding lat-lon axes.

We have added lat/lon axes to this figure. These images represent qualitative inundation surface categories, though, so scale bars are not relevant.

R2-2) Bathtub model oversimplifies impact of SLR on coastlines.

A short paragraph (line 232) has been added to address this. It's clear modeling of geomorphological processes that affect coastline position is beyond the scope of this paper, but it is worth noting that the impact of SLR on how those processes operate is unknown and could also play a major role in affecting coastal population vulnerability (and could work to increase population vulnerability in some areas while decreasing it in others), particularly in non-urban areas.

A note to this effect has been added.

Reviewer #2 (Remarks to the Author):

This new version is only slightly revised and changed with respect to the original submission, despite some major concerns pointed out by three different reviewers. My recommendation therefore remains the same: the manuscript should still undergo a revision before being published.

One major issue in the previous version was the poor treatment and discussion of the uncertainties in CoastalDEM (only uncertainties associated to mean sea level rise were accounted for, but not uncertainties in the DEM product). This is a point that has been raised by the three reviewers. To address this, the authors have performed an additional exercise to evaluate how uncertainties in CoastalDEM are translated into uncertainties in exposure, using Australia as a case study. I see two problems with the response: firstly, it is now hidden in a file of the S.I. while, due to its relevance, it should be more central to the discussion and highlighted in the main text. Secondly, and perhaps the reason why it is in S.I., I cannot draw a main conclusion from the sensitivity tests. Does it serve to estimate an uncertainty range in population exposure associated to the use of this DEM?

The use of multiple spatial resolutions in the DEM fields is clearly a limitation. I do not think that this is the way it should be done; instead, I think that the DEM fields constructed using uncertainties should have the same resolution as the original CoastalDEM and should be spatially correlated. This is a non-trivial task and I am not suggesting that the authors follow this approach. An alternative, easier way to bound the uncertainty in the population exposure associated to the errors in CoastalDEM would be to increase/decrease the DEM by the same relative amount everywhere around the Australian coast: for example, +/- 50% of the RMSE at every pixel.

Upon request, the reviewer kindly provided this clarification:

My concern is that the reader still does not have a notion of the uncertainty associated to the DEM. My suggestion to do so is the following: use only the original 3-arcsec resolution and generate two error surfaces that correspond to +/-1-sigma values of the uncertainty at each pixel. If the errors are correlated so will be their relative uncertainties. Let's imagine two consecutive pixels with A and B altitudes and eA and eB errors; the error surface would be $A+0,6*eA$ and $B+0,6*eB$ for the upper bound and similarly for the lower bound. In this way, I think the spatial correlations would remain.

There is, I think, another easy way to handle this. Generate, at the same spatial resolution, ~1000 error surfaces assuming Gaussian distributions around the mean value and with a standard deviation given by the error at each pixel. Then, average all the error surfaces and compute the sigma values at each pixel around the mean, which will also keep the spatial correlation (essentially you will be filtering out the undesirable spatial noise).

Both alternatives are possible and, although not perfect, would provide a measure of the uncertainty associated to the DEM in a region as Australia where the errors at each pixel can be estimated by comparison with high-resolution data.

We sincerely appreciate the thought the reviewer put into this suggestion, but we strongly believe that either such approach would not achieve a better understanding of global-scale exposure sensitivity than what we have done in this revised manuscript. We also believe the reviewer may not have appreciated that all of our simulated DEMs retained CoastalDEM's original 3-arcsecond resolution; only the added error surfaces used coarser resolutions. We accept responsibility for unclear writing in this regard in our last submission, and have strived to make improvements for this one.

Regarding the reviewer's former suggestion, this approach, as we understand it, would not provide insight on how different scales of autocorrelation (e.g., random noise, cities, striping, etc.) influence sensitivity. Further, any results would reflect detailed patterns of error specific to Australia that may or may not be indicative of errors elsewhere. On a related but distinct note,

this method is impossible to apply at global scale, since we do not have a known error surface everywhere.

The second suggestion, as we understand it, would also not be possible to apply globally, for the same reason. We further note that the mean vertical error in Australia is virtually zero. If an error value at a pixel is then randomly generated using a Gaussian distribution, using its error compared to lidar as the standard deviation and with zero mean, then when we average all the error surfaces together, the pixel values will cancel each other out and we will be left with a surface of essentially zero error.

It is possible we do not fully understand Reviewer 2's suggestions. However, we believe we understand enough to appreciate that they would not be applicable globally, whereas we have found an approach that is. Furthermore, we are hopeful that with our expanded analysis and discussion, Reviewer 2 will find to his or her satisfaction that we provide readers with much more insight into the potential sensitivity of our results to DEM error, the ultimate goal of his/her original and well-taken comments. (This sort of analysis is essentially absent from the past literature on global exposure to sea-level rise, to our knowledge, but we understand the interest in it in our case because of the importance we claim for using new elevation data. We also note that our study is unprecedented simply in providing exposure estimates using four different global elevation datasets, as well as comparing these to results based on ground truth (lidar) where feasible.)

As discussed in a response to Reviewer 1, Monte Carlo simulations are very frequently used in the literature to understand the sensitivity of exposure estimates to DEM error, and our approach makes straightforward changes to traditional techniques to gain further insight on the effect of autocorrelation.

Regarding the figures:

- Modification in Figure 1: I suggested to include the present-day coastlines in Figure 1 as the locations are extremely difficult to recognise without this information. I think no one could actually identify the coast of Tampa using only flooded Lidar data shown in the first column of the figure. I understand from the authors' responses that this may be a computationally costly action, but I believe this is needed to properly interpret the figure.

As discussed above in a response to Reviewer #1, we have removed (what was previously) Figure 1 from the manuscript. The figure comparing inundation surfaces already adequately illustrate how projected coastlines according to CoastalDEM and SRTM compare to the present-day versions.

- Figure 5: there has been an improvement in the quality, but I still find the caption insufficient to interpret what is shown here. Why are now values centred at -0.5? what is

exactly the y-axis? In the former version I would have said that it represents a percentage, but now I am unsure given its average value.

There was a bug in the code that caused the y labels to be wrong-- this has been fixed and values are centered about 0. The y axis is the relative difference between projected population exposure according to lidar and each global DEM. We have edited the caption to make this clearer.

- Line 84: this sentence needs to be rewritten: if there is permanent flooding below MHHW, then is not due to ESL but to mean sea level rise. Only temporary flooding is associated to ESL.

This has been resolved.

It would have been very helpful that the authors indicated where (e.g. page, lines) the new modified text has been included, or, even better, would have provided a file with tracked changes.

We have attached a file with tracked changes to this resubmission. Please see our general remarks to all reviewers, above, for a description of the modest limitations of our tracking.

Reviewer #3 (Remarks to the Author):

General remarks

The updated manuscript is a substantial improvement over the previous version, although there are still some open issues that were raised in the previous round, but have not yet been fully addressed in my opinion.

The major weakness of this paper still forms the estimates of the uncertainty. Although I can see that a full assessment of all sources of uncertainty is not feasible to perform, the readers are still left in the dark on how to assess its uncertainty on a global scale, at least after reading the supplement, I still don't have an idea of how to interpret the numbers in the main text. The authors have added a case study in Australia, but some basic guidelines or a first-order estimate how these local results translate into other regions and the global-mean case is still missing.

We understand the reviewer's concerns, and as discussed above, we have made substantial improvements to this sensitivity analysis, performing it not just in Australia, but every other coastal country, as well as the world as a whole.

Furthermore, I still don't understand how the K14/K17 projections are interpolated on the whole coastline. These projections are not gridded but computed at a set of tide-gauge locations. This is especially important given the background rates that are included in K14 (Figure 6c), which are derived from local tide-gauge data and not from gridded estimates. Are these local values interpolated along the coastlines? Do you also interpolate the values from Figure 6c? This is especially important, given that this term contains large scale (e.g. GIA) and local (e.g. subsidence due to groundwater pumping) effects, which are not always suitable for interpolation. For reproducibility, the followed procedure should be part of the methods section.

These projections are, in fact, gridded at 2-degree horizontal resolution, though also include data specifically at the tide-gauge locations as well. The K14 projections originally were not so in the 2014 paper, but these have been updated in the Kopp 2017 paper (section 2.4). In our analysis, intermediate values between the 2 degree cells and/or tide-gauges are linearly interpolated, and we edited the text to reflect this linear interpolation.

Finally, in the previous review, I asked about the number of people currently living in risky areas. While this number has been added, I wonder where does the number '280 million' come from? From coastalDEM? In that respect, is there a difference in the number of people who are now not inundated and will be in the future between SRTM and coastalDEM? Both the present-day and future inundation are biased low in SRTM I guess. Another option would be to express all numbers as the change in number of affected people relative to the present-day baseline. Furthermore, this high number of people already under the annual flood level would put the large numbers of people affected in better perspective. The maximum number of people at risk is 640 million, which is just more than double the number of people presently at risk. I therefore strongly suggest to make this number of 280 million very clear from the beginning, to avoid the risks that this paper generates news headlines that are not justified. Finally, how many people nowadays live below MHHW? Many coastal cities in Europe and Asia have large parts protected by seawalls.

The reviewer makes good points, and we have significantly extended the discussion on this topic in this revision, beginning with the abstract, where headlines are made. In the results section, present-day totals (both below MHHW and RL1) for CoastalDEM and SRTM are reported first, before any other projection. Further, later in the paper we discuss potential sources of error for these present-day numbers (likely to be in areas currently protected by infrastructure, such as levees, or due to vertical error in the global MHHW surface, or errors in the population data).

However, we very much disagree that all numbers should be presented as the change relative to present-day values. First of all, today's protective infrastructure may be insufficient against future sea levels in many locations. On the other hand, today's protective infrastructure may

effectively protect populations that our analysis indicates to be exposed only in the future. Therefore, in our view, taking a difference does not clarify, but rather adds an unnecessary step and threatens to confuse. However results are presented, there is unavoidable ambiguity around which areas may be effectively protected by today's -- or tomorrow's -- defensive infrastructure. We explicitly address these concerns in our discussion, and hope our discussion will satisfy readers and the reviewer.

Line-by-line remarks

L7: A recent paper by Gregory et al. (2019) suggests a standardization of many sea-level related terms. One of them is 'Extreme Coastal Water Level' instead of Extreme Sea Level. I'd like to encourage the authors the consistently use the terminology proposed by Gregory et al (2019)

We have edited the manuscript to use this terminology.

L11; 'More than 200 million'. From Table 1, I get a value of 'more than 170 million', which is less than 3 times the value of 65 million from SRTM.

We have updated the numbers and the corresponding text.

L14: 360M mid-century: This number is not discussed in the main text. I suggest to remove it.

We have added this number to the main text.

L45: Square mile: Please use SI units.

This has been edited.

L95/ Table 1: Where do these uncertainties come from? It looks like they're based on the projected uncertainties only, but that should be made explicit.

We have edited the manuscript to make this explicit.

L105FF: This is a good example on why the baseline would matter. How high are these numbers today, and do they depend on SRTM vs coastalDEM?

This has been resolved with the inclusion of all present-day estimates.

L132-L137. I already raised this point in the previous section, but I still don't see the added value of these statements, as these effects are already known, and the probability of these effects are not further quantified in this manuscript. This is a scientific paper after all, and not a textbook or press release.

We have moved these comments from the results section to the end of the discussion, where they more appropriately belong. We feel it is relevant and appropriate to provide some indications of the broader societal importance and context.

L248-L249: These statements only hold for regions and communities that have based their expectations on SRTM or other unreliable DEMs. Is this actually the case? I suspect that many places either have used a high-resolution estimate (developed countries) or do not have any estimate at all (under-developed countries). About the immigration pressure: this is not trivial either, see for example Eseban et al, 2018, who show that people tend to stand their ground. I again suggest to stick to the computed numbers in the conclusions.

This statement is true for places without high-quality elevation data, after one small edit we make in light of this comment, inserting “may be” before “currently anticipated,” to account for places lacking estimates, or awareness of them, altogether. That said, a number of previous studies have performed a similar global-scale vulnerability analysis as we are here, and included national level results, but using considerably less accurate elevation data. The results here are a major improvement that we hope will better inform developing nations on the coastal threats they may soon be facing; and as we note, the great majority of world population at risk is on the developing continent of Asia. Especially for smaller countries and areas, though, better elevation data remains an extremely important goal as such places evaluate coastal threats and adaptation options.

As for the immigration pressure, we understand that modelling projected migration is currently extremely difficult, which is why we do not attempt it here. However, while it may be true that people tend to stand their ground, the point here is that without plans to improve coastal protective infrastructure, our results suggest that more places than we originally thought will be threatened by nearly or totally unlivable conditions that leave residents with no choice but to leave.

Reviewers' comments:

Reviewer #1 (Remarks to the Author):

This is a significant revision that appears to have addressed my concerns from the earlier manuscript versions. A fresh reading indicates a clear contribution to the SLR impacts literature. The paper is better organized, grounds its findings in the literature and the analysis, and communicates those findings effectively through the graphics and tables.

I am not a fan of the simulation method used to evaluate the impact of error, but I appreciate the decision to subject the entire coastal areas of SRTM to an error propagation assessment. The results provide reasonable confidence intervals on the number of (modern-day) people affected, given the error properties of CoastalDEM. I think the approach holds up.

I'd like to see this work published, and think that it's ready.

Reviewer #2 (Remarks to the Author):

I appreciate the effort the authors have made to clarify my major concern on the uncertainty bounds. I totally understand their response and mostly agree with their comments.

My first suggestion was to perform the analyses in Australia only, and thus the reference I made to the errors at every pixel; but I understand it is not necessarily representative of other regions and certainly feasible at the global scale, as it has been now done. Regarding my second suggested analysis I do not agree that the error surfaces would cancel each other out. Nevertheless, it is irrelevant now considering the new explanations provided in the manuscript.

I think that the new section on sensitivity analyses is enlightening. The procedure is much clearer now and it is, in my opinion, valuable for the reader that it is placed in the main text. Many points have been clarified, such as the use of the original spatial resolution in every error-block size, that I misunderstood in the former version. I also appreciate the extension of the sensitivity analysis to the global coastlines instead to the Australian case. In this respect, I wonder whether the Australian RMSE that has been used everywhere as the standard deviation to build the error surfaces is globally representative. The authors may want to consider a comment on this point.

My recommendation is that the manuscript is now in good shape for publication.

Reviewer #3 (Remarks to the Author):

Review of 'New Elevation Data Triple Estimates of Global Vulnerability to Sea-level Rise and Coastal Flooding' by Scott A. Kulp and Benjamin H. Strauss, round 3

The authors have made some significant changes to the manuscript, and many of my concerns have been addressed, especially the uncertainty analysis, which is very helpful for understanding the results. However, I'm still concerned with the difference between the total number of people exposed and the increase of the number of people exposed. In the rebuttal letter the authors argue that they don't want to present the exposure relative to the present-day exposure, because it is unknown whether the present-day protective infrastructure will be sufficient for future MSL and ECWL changes. As far as I see, the new DEM basically just results in more people living in low-lying areas, and therefore, the number of people living in at-risk areas both now and under various sea-level rise scenarios is higher. While this fact is very interesting and worthy of publication, I'd still be reluctant to focus on the comparison of future exposure between both product without taking present-day exposure into account. I'd argue that the multiplication factor of the number of people at risk, which is even in the title, could easily be prone to wrong interpretation. An example of this mis-interpretation already occurs in the manuscript, namely the statement on line 106:

In the case of Antarctic instability, 300 (270-340) million people may be living on land today that could become vulnerable to an annual flood event by as early as mid-century, rising to as many as 500 (390-640) million by 2100.

When I read this sentence, I get the impression that 300 million people could become vulnerable to flooding due to sea-level rise. However, of these 300 million people, 250 million people already live in an area that is vulnerable to flooding, and without any knowledge on how the risk for these people changes under sea-level rise, the meaning of this number is ambiguous at least. Changing this sentence to something like

'In the case of Antarctic instability, the extra land that could become subject to an annual flood event by as early as mid-century is currently home to 50 (20-90) million people, rising to as many as 250 million people by 2100.'

resolves this ambiguity, while the main message of the paper (improved DEM shows more people are vulnerable to flooding) is still clear. Therefore, I'd strongly recommend to change all the numbers to reflect the increase w.r.t. the present-day situation. These numbers also show that with the more accurate DEM many more people will be at risk than previously thought, while it avoids any confusion.

Minor remarks

- The sea-level projections: as I now understand that all projections are based a combination of the gridded projections and the projections at the tide-gauge locations. Since the tide-gauge projections contain a local term due to residual processes (K14 figure 6c), which must be absent in the gridded projections, as the spatial extent of these residual processes is unknown, I wonder how you do this interpolation. A simple solution would be just to use the gridded product, which avoids this issue altogether.

- A recurring discussion during this review process is on whether to refer to potential consequences of sea-level rise (line 362ff). I still have lukewarm feelings about this, as it is out of scope for this paper and it also contradicts with the earlier statement on line 291: Results should therefore not be taken as projected impacts. There may be a role for the editor to judge on this issue.

Once again, we would like to thank the editor and reviewers for their time and thoughtful comments. We have made several changes to the manuscript that improve it even further, and we hope that all agree that it is ready for publication. As with the previous submission, we have attached both a clean version of the manuscript, as well as one with changes marked.

Point-by-point responses to the reviewers, where relevant to requested changes to the manuscript, follow.

Reviewer #2 (Remarks to the Author):

I wonder whether the Australian RMSE that has been used everywhere as the standard deviation to build the error surfaces is globally representative. The authors may want to consider a comment on this point.

We have added a short comment reflecting on this point to the manuscript.

Reviewer #3 (Remarks to the Author):

The authors have made some significant changes to the manuscript, and many of my concerns have been addressed, especially the uncertainty analysis, which is very helpful for understanding the results. However, I'm still concerned with the difference between the total number of people exposed and the increase of the number of people exposed. In the rebuttal letter the authors argue that they don't want to present the exposure relative to the present-day exposure, because it is unknown whether the present-day protective infrastructure will be sufficient for future MSL and ECWL changes. As far as I see, the new DEM basically just results in more people living in low-lying areas, and therefore, the number of people living in at-risk areas both now and under various sea-level rise scenarios is higher. While this fact is very interesting and worthy of publication, I'd still be reluctant to focus on the comparison of future exposure between both product without taking present-day exposure into account. I'd argue that the multiplication factor of the number of people at risk, which is even in the title, could easily be prone to wrong interpretation. An example of this mis-interpretation already occurs in the manuscript, namely the statement on line 106:

In the case of Antarctic instability, 300 (270-340) million people may be living on land today that could become vulnerable to an annual flood event by as early as mid-century, rising to as many as 500 (390-640) million by 2100.

When I read this sentence, I get the impression that 300 million people could become vulnerable to flooding due to sea-level rise. However, of these 300 million people, 250 million people already live in an area that is vulnerable to flooding, and without any knowledge on how the risk for these people changes under sea-level rise, the meaning of this number is ambiguous at least. Changing this sentence to something like

‘In the case of Antarctic instability, the extra land that could become subject to an annual flood event by as early as mid-century is currently home to 50 (20-90) million people, rising to as many as 250 million people by 2100.’

resolves this ambiguity, while the main message of the paper (improved DEM shows more people are vulnerable to flooding) is still clear. Therefore, I’d strongly recommend to change all the numbers to reflect the increase w.r.t. the present-day situation. These numbers also show that with the more accurate DEM many more people will be at risk than previously thought, while it avoids any confusion.

We appreciate the reviewer’s concern over using the total exposure statistics alone. We have edited the manuscript to prominently include results based on marginal exposure, and to highlight and discuss the merits and differences between the two approaches. We have also corrected the sentence the reviewer highlighted, and feel the manuscript is considerably clarified and improved by these changes. At the same time, we very much disagree that total exposure results should be disregarded and removed. This metric is nearly universal in the literature on exposure to sea level rise and coastal flooding. More importantly, we believe that reporting marginal differences alone would almost certainly under-represent the risk coastal communities will face in the coming decades.

The reviewer is correct that our main conclusion would be sustained, even if focusing on just marginal exposure differences. However, the comparison of results between CoastalDEM and SRTM would be complicated to its detriment, with greatly disproportionate influence given to results differences at 0 meters. Many important comparisons and figures in the manuscript would become tortured in their interpretation, if not moot, such as the sensitivity analysis. The full consequence of switching to a main or exclusive focus on marginal exposure -- vs. total exposure -- would be a major overhaul of the entire manuscript, in our view, and with loss in clarity.

We further justify our inclusion of total exposure metrics in our revisions to the manuscript. Again, we appreciate the reviewer’s concern, and we hope we have addressed it enough for her or his satisfaction.

- The sea-level projections: as I now understand that all projections are based a combination of the gridded projections and the projections at the tide-gauge locations. Since the tide-gauge projections contain a local term due to residual processes (K14 figure 6c), which must be absent in the gridded projections, as the spatial extent of these residual processes is unknown, I wonder how you do this interpolation. A simple solution would be just to use the gridded product, which avoids this issue altogether.

We have edited to manuscript to more explicitly describe how updated K14 and K17 grids are computed in the Kopp et. al 2017 paper, and in our methods section, we also describe how we interpolate the projection between grid points and tide stations. The main point is that the grid points incorporate information from the tide stations via a Gaussian process model, and so take into account local nonclimatic processes, and are broadly consistent with the tide station results. We suspect that that should satisfy the reviewer.

Per Bob Kopp, lead author of both papers, with whom we consulted: “I don’t see a good rationale for excluding the tide gauges, where there are more precise constraints on the geological background rate.”

Rerunning the analysis using a different projection surface would be an immense time and computational cost for us, and would actively reduce the accuracy and quality of the projections and our results. We therefore very much disagree with the suggestion to use the gridded results alone.

- A recurring discussion during this review process is on whether to refer to potential consequences of sea-level rise (line 362ff). I still have lukewarm feelings about this, as it is out of scope for this paper and it also contradicts with the earlier statement on line 291: Results should therefore not be taken as projected impacts. There may be a role for the editor to judge on this issue.

We agree to defer to the editor on this matter.

REVIEWERS' COMMENTS:

Reviewer #2 (Remarks to the Author):

I am satisfied with the last amendment made to the manuscript and I think it deserves publication in its present form.

Reviewer #3 (Remarks to the Author):

Review of 'New Elevation Data Triple Estimates of Global Vulnerability to Sea-level Rise and Coastal Flooding' by Scott A. Kulp and Benjamin H. Strauss, round 4

The changes that the authors made to highlight the marginal exposures is in my opinion a satisfactory improvement of the paper and I recommend publication of the current version.

The only unresolved point of discussion is about potential impacts of this analysis, starting on lines 392ff. I leave the judgement on whether this paragraph is appropriate to the editor.